# *Caenorhabditis elegans* RMI2 functional homolog-2 (RMIF-2) and RMI1 (RMH-1) have both overlapping and distinct meiotic functions within the BTR complex

**Maria Velkova**[1], **Nicola Silva**[1¤], **Maria Rosaria Dello Stritto**[1], **Alexander Schleiffer**[2,3], **Pierre Barraud**[4], **Markus Hartl**[5], **Verena Jantsch**[1]*

**1** Department of Chromosome Biology, Max Perutz Labs, Vienna BioCenter, Vienna, Austria, **2** Research Institute of Molecular Pathology, Campus Vienna BioCenter, Vienna 1, Vienna, Austria, **3** Institute of Molecular Biotechnology, Campus Vienna BioCenter, Vienna, Austria, **4** Expression Génétique Microbienne, UMR 8261, Centre national de la recherche scientifique, Université de Paris, Institut de Biologie Physico-Chimique, Paris, France, **5** Mass Spectrometry Facility, Max Perutz Labs, Vienna BioCenter, Vienna, Austria

¤ Current address: Department of Biology, Faculty of Medicine, Masaryk University, Brno, Czech Republic
* verena.jantsch@univie.ac.at

**Data Availability Statement:** All relevant data are within the manuscript and its Supporting Information files.

## Abstract

Homologous recombination is a high-fidelity repair pathway for DNA double-strand breaks employed during both mitotic and meiotic cell divisions. Such repair can lead to genetic exchange, originating from crossover (CO) generation. In mitosis, COs are suppressed to prevent sister chromatid exchange. Here, the BTR complex, consisting of the Bloom helicase (HIM-6 in worms), topoisomerase 3 (TOP-3), and the RMI1 (RMH-1 and RMH-2) and RMI2 scaffolding proteins, is essential for dismantling joint DNA molecules to form non-crossovers (NCOs) via decatenation. In contrast, in meiosis COs are essential for accurate chromosome segregation and the BTR complex plays distinct roles in CO and NCO generation at different steps in meiotic recombination. RMI2 stabilizes the RMI1 scaffolding protein, and lack of RMI2 in mitosis leads to elevated sister chromatid exchange, as observed upon RMI1 knockdown. However, much less is known about the involvement of RMI2 in meiotic recombination. So far, RMI2 homologs have been found in vertebrates and plants, but not in lower organisms such as *Drosophila*, yeast, or worms. We report the identification of the *Caenorhabditis elegans* functional homolog of RMI2, which we named RMIF-2. The protein shows a dynamic localization pattern to recombination foci during meiotic prophase I and concentration into recombination foci is mutually dependent on other BTR complex proteins. Comparative analysis of the *rmif-2* and *rmh-1* phenotypes revealed numerous commonalities, including in regulating CO formation and directing COs toward chromosome arms. Surprisingly, the prevalence of heterologous recombination was several fold lower in the *rmif-2* mutant, suggesting that RMIF-2 may be dispensable or less strictly required for some BTR complex-mediated activities during meiosis.

**Funding:** The work was funded by FWF (Fonds zur Förderung der wissenschaftlichen Forschung, https://www.fwf.ac.at/en/) projects P 31275-B28 (to VJ) and F-34 (to MH). MV and MRDS were supported by the doctoral school "Chromosome Dynamics", FWF (Fonds zur Förderung der wissenschaftlichen Forschung, https://www.fwf.ac.at/en/) project W1238. MV is funded by a uni:doc fellowship from the University of Vienna. NS's laboratory is funded by the Grant Agency of the Czech Republic, https://gacr.cz/en/, (GA20–08819S) and a Start-Up grant from the Department of Biology, Masaryk University. Boehringer Ingelheim and the Austrian Academy of Sciences support AS. PB acknowledges financial support from the Centre national de la recherche scientifique. Some worm strains were provided by the Caenorhabditis Genetics Center, which is funded by the NIH Office of Research Infrastructure Programs (P40OD010440), https://orip.nih.gov/. The funders had no role in study design, data collection and analysis, decision to publish, or preparation of the manuscript.

**Competing interests:** The authors have declared that no competing interests exist.

## Author summary

Bloom syndrome is caused by mutations in proteins of the BTR complex (consisting of the Bloom helicase, topoisomerase 3, and the RMI1 and RMI2 scaffolding proteins) and the clinical characteristics are growth deficiency, short stature, skin photosensitivity, and increased cancer predisposition. At the cellular level, characteristic features are the presence of increased sister chromatid exchange on chromosomes; unresolved DNA recombination intermediates that eventually cause genome instability; and erroneous DNA repair by heterologous recombination (recombination between non-identical sequences, extremely rare in wild type animals), which can trigger translocations and chromosomal rearrangements. Identification of the *Caenorhabditis elegans* ortholog of RMI2 (called RMIF-2) allowed us to compare heterologous recombination in the germline of mutants of various BTR complex proteins. The heterologous recombination rate was several fold lower in *rmif-2* mutants than in mutants of *rmh-1* and *him-6* (worm homologs of RMI1 and the Bloom helicase, respectively). Nevertheless, many phenotypic features point at RMIF-2 working together with RMH-1. If these germline functions of RMI2/RMIF-2 are conserved in humans, this might mean that individuals with RMI2 mutations have a lower risk of translocations and genome rearrangements than those with mutations in the other BTR complex genes.

## Introduction

Damage-induced DNA double-strand breaks (DSBs) pose a threat to genome integrity. High-fidelity repair via homologous recombination (HR) is employed during both mitotic and meiotic cell cycles. It involves the generation of 3′ overhang ends by DNA resection and their stabilization by the single-stranded DNA-binding protein RPA (replication protein A) (RPA-1 in worms). RPA-1 is subsequently exchanged with the RAD-51 recombinase to allow invasion of a homologous DNA strand, giving rise to a D-loop intermediate structure. After DNA synthesis and second-end capture, DNA joint molecules are generated. These can be processed to produce crossovers (COs), which result in the reciprocal exchange of large regions of chromosomes [1]. In meiosis, where Spo11-mediated DSBs are induced via a highly regulated program, crossing-over and cohesion establish a physical tether between homologous chromosomes, which greatly aids their correct segregation in meiotic anaphase I and drives genetic variability. In meiosis, one chromatid of the homologous chromosome is preferentially used as a repair template for HR. Joint DNA molecules must disengage in order to segregate, and this is achieved by redundant endonucleases (called resolvases) and the BTR complex [2]. Depending on the orientation of the resolvase-induced cut, the outcome is a CO or non-crossover (NCO) product. To ensure at least one CO per chromosome pair, excess DSBs are introduced and those that do not form the CO are repaired to form NCOs [2].

The BTR complex dismantles joint DNA molecules via its decatenation activity, which has been reconstituted *in vitro* [3]. In decatenation, strand passage is achieved via cutting one DNA strand and then resealing the DNA break. Following HR in mitotically dividing cells, the BTR complex mostly mediates the NCO outcome since COs can have detrimental effects such as loss of heterozygosity [3]. For example, loss of heterozygosity of a tumor suppressor gene can lead to cancer development. Patients with mutations in components of the BTR complex show elevated rates of sister chromatid exchange and aberrant chromosomes [4,5].

In mammals, the BTR complex consists of Bloom helicase, topoisomerase, and the RMI1 and RMI2 scaffolding proteins. In worms, the respective homologs are HIM-6, TOP-3, and

RMH-1 and RMH-2 (both RMH proteins are RMI1 homologs—an RMI2 homolog has not been identified). Structural and biochemical analysis of RMI1 suggest its involvement in the strand passage–tyrosine transesterification reaction mediated by the topoisomerase. Based on these activities combined with the DNA unwinding activity of the helicase, the BTR complex has important roles in DNA metabolism [6,7].

Detailed analyses of the homologous yeast STR complex (Sgs1 helicase–topoisomerase–Rmi1) have revealed that its meiotic functions include an important role in D-loop reversion *in vivo* [8,9], which has also been shown *in vitro* [10]. D-loop reversion prevents the generation of complex multi-joint molecules; in the absence of STR activity, multi-joint molecules can only be resolved by non-canonical resolvases, which generate a mix of COs and NCOs (i.e. additional COs are formed). It was also observed that Top3-Rmi1 form a sub complex that limits the accumulation of toxic recombination intermediates. Loss of function of both Rmi1 and topoisomerase 3 leads to meiotic catastrophe, due to persistent joint molecules that are resistant to cleavage by resolvases. It is conceivable that these DNA structures represent extended D-loops involving homologous and/or heterologous chromosomes or other branched structures (for a review, see [11]). The *Caenorhabditis elegans* Bloom ortholog HIM-6, similar to BRC-1/BRCA1, suppresses heterologous recombination in the germline, which could lead to translocations and genome rearrangements [12].

Overall, we know now from many model systems that the BTR complex plays separable roles in CO and NCO formation during meiosis and governs the number and placement of CO sites along the chromosomes [8,9,13–25]. In the *C. elegans* model system, pro-CO activity is particularly obvious in mutants, due to the presence of univalent chromosomes at diakinesis [19,22,26].

In mammalian cells, RMI2 was identified as an RMI1- or Bloom-interacting protein [27,28]. Like RMI1, it contains a characteristic OB-fold domain (OB, oligonucleotide/oligosaccharide binding), and RMI1 and RMI2 interact via their OB domains (OB2 in RMI1 and OB3 in RMI2). RMI2 is required to stabilize the other members of the complex within recombination foci, and is also suggested to function in governing post-translation modifications of other complex members [27]. Upon RMI2 depletion, elevated sister chromatid exchange and chromosome aberrations have been observed [28].

So far, RMI2 orthologs have not been identified in yeast, *Drosophila*, or *C. elegans*. Here we report the identification of a novel *C. elegans* protein encoded by the open reading frame Y104H12D.4, which we found in RMH-1-containing protein complexes in the germline. Similar to RMI2 proteins, Y104H12D.4 contains an OB-fold domain. Based on its ability to stabilize RMH-1 (worm RMI1) and concentrate HIM-6 (Bloom helicase) and topoisomerase 3 into recombination foci, it qualifies as a functional homolog of RMI2. Thus, we named the protein RMI2 functional homolog-2 (RMIF-2). Similar to the other BTR complex proteins, RMIF-2 displays a dynamic localization pattern in recombination foci. Nevertheless, our detailed analysis of germline recombination revealed marked differences between *rmh-1* and *rmif-2* mutants, indicating that *rmif-2* functions not just as RMH-1 stabilizer for all its activities in the germline as it is observed in mammalian cells. Our data suggest, that without RMIF-2 RMH-1 can function in some of the BTR related meiotic activities.

## Results

### Identification of Y104H12D.4 as an interaction partner of RMH-1

To purify RMH-1 meiotic interaction partners, we tagged the endogenous *rmh-1* locus with a 5′ HA-degron-tag using CRISPR/Cas9. The tagged line has normal hatching rates (comparable to those of the GFP-tagged line, which we published previously [19]), indicating its full functionality. Biochemical fractionation [29] and western blot analysis showed that the RMH-1

**Table 1. Interacting proteins of RMH-1 as determined by affinity purification mass spectrometry.**

| Gene names | Unique peptides | PSM | | | | | | log$_2$ ratio RMH-1 / CTRL | LIMMA p-value | LIMMA adj. p-value |
|---|---|---|---|---|---|---|---|---|---|---|
| | | RMH-1 bait | | | Control | | | | | |
| | | r1 | r2 | r3 | r1 | r2 | r3 | | | |
| *rmh-1* | 45 | 69 | 79 | 19 | 0 | 0 | 0 | 10.1 | 1.2E-05 | 0.003 |
| *top-3* | 37 | 22 | 69 | 28 | 0 | 0 | 0 | 8.7 | 4.4E-06 | 0.002 |
| *him-6* | 24 | 11 | 28 | 6 | 0 | 0 | 0 | 7.8 | 3.5E-06 | 0.002 |
| *rmif-2* | 5 | 2 | 4 | 2 | 0 | 0 | 0 | 6.0 | 3.9E-05 | 0.006 |
| *rpa-1* | 6 | 4 | 5 | 3 | 0 | 0 | 3 | 3.7 | 0.011 | 0.573 |

Expected interactors were TOP-3, HIM-6, and RPA-1. RMIF-2 (Y104H12D.4) was identified as a novel RMH-1-interacting protein. Peptide spectrum matches (PSM) indicate how often peptides of a given protein were identified from spectra. The log$_2$ ratio is computed from protein intensities (as a measure of protein abundance), which were also used for the statistical analysis (LIMMA).

protein is enriched in both nuclear soluble and insoluble fractions. To identify RMH-1 interactors, we used pooled nuclear fractions in immunoprecipitation experiments followed by mass spectrometry analysis. In triplicate experiments, several interaction candidates (Table 1) were found to be enriched over control levels, including RPA-1 and other members of the BTR complex: HIM-6 and TOP-3. The novel open reading frame Y104H12D.4 (RMIF-2) was also identified as an RMH-1 interactor.

Despite the presence of RMI2 family members in other nematodes, the *C. elegans* proteome does not contain a Pfam RMI2 hit (see Materials and Methods for details of the bioinformatics analysis). Of the co-purified proteins, RMIF-2 was the only candidate with a predicted OB-fold domain to be identified with methods to find remote homologs (Fig 1A). A ribbon diagram of the human RMI core complex and a model of the putative *C. elegans* RMIF-2 OB-fold can be found in S1 Fig. In this report, we will present evidence that RMIF-2 is a true functional homolog of RMI2.

To confirm the interaction between RMH-1 and RMIF-2, we generated a functional *rmif-2::3xflag* tagged line (Table 2), which we combined with *ha::rmh-1* [30]. We used this strain for reciprocal co-immunoprecipitation experiments. Western blot analysis of FLAG pull downs revealed robust co-immunoprecipitation of HA::RMH-1, confirming that these two proteins form a complex *in vivo* (Fig 1B).

## RMIF-2 displays a dynamic localization throughout pachynema

To gain insight into RMIF-2 expression and subcellular localization during meiotic prophase I, we added a 3′ HA-tag to the *rmif-2* locus. The tagged line displayed wild type (WT) hatching rates and brood size, indicating that the strain is functional (Table 2). RMIF-2::HA localized into dynamic foci throughout pachynema (Fig 1C). The protein started to accumulate as foci in early pachytene nuclei (mean 5.6 (± 5.8 SD) foci per nucleus; Fig 1D). The number of RMIF-2 foci peaked in mid pachynema (range 2–24 foci per nucleus; mean 9.0 (± 4.6 SD)). Most late pachytene cells contained five RMIF-2::HA foci (range 0–7 foci per nucleus). This dynamic localization pattern strongly resembles that of RMH-1 [19]; in addition, RMH-1 and RMIF-2 foci extensively co-localized (Fig 1D and 1E).

## *rmif-2* is required for meiotic recombination, robust chiasma formation, and chromosome segregation in meiosis

To further analyze the meiotic function of RMIF-2 and its functional co-operation with other BTR complex members, we deleted the entire *rmif-2* locus to generate a null allele. The

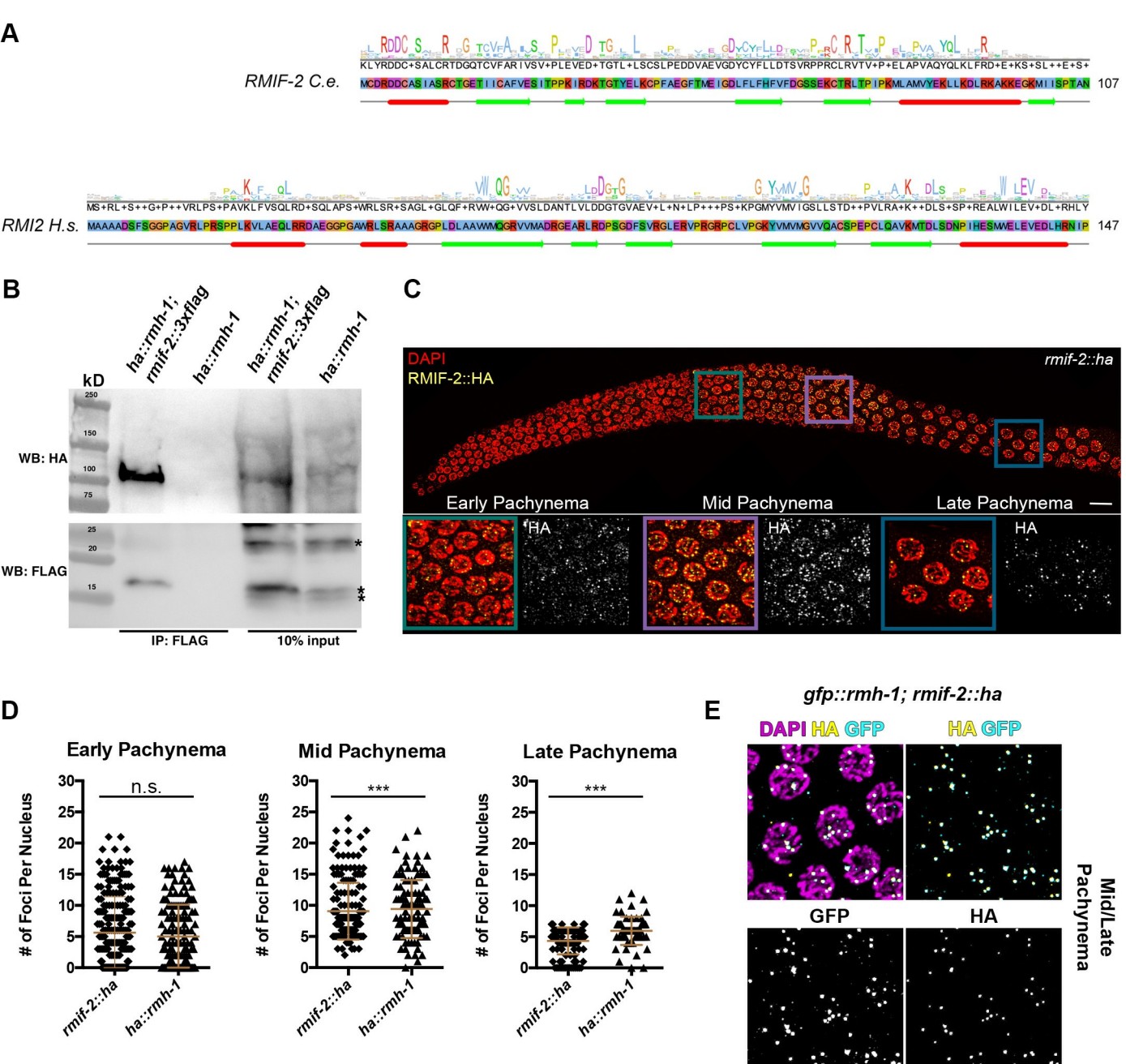

**Fig 1. RMIF-2 as a functional homolog of RMI2.** (A) A conservation histogram and consensus sequence (top lines), primary sequence (middle) and a secondary structure prediction (bottom line) of *Caenorhabditis elegans* RMIF-2 (UniProt accession Q8MXU4) and *Homo sapiens* RMI2 (Q96E14). In the case of RMIF-2, the conservation histogram and the consensus sequence are based on an alignment of nematode orthologs, and for RMI2 a wide selection of eukaryotic orthologs was used, including animal and plant sequences. Sequence letters were highlighted in the ClustalX color scheme to indicate amino acids with similar physicochemical properties. Secondary structure elements were predicted (Jpred), where the helices are marked as red tubes, and sheets as green arrows (JNETPSSM), [59]. Both families share the sequential arrangement of a five-stranded beta sheet and a c-terminal alpha helix. (B) Western blot analysis of FLAG pull downs revealed robust co-immunoprecipitation of HA::RMH-1 and RMIF-2::3×FLAG. *ha::rmh-1* worms were used as the negative control. The predicted size of RMIF-2::3×FLAG is 16 kD and HA::RMH-1 109 kD. IP, immunoprecipitation; WB, western blot. Asterisks indicate unspecific bands. (C) A Representative image of RMIF-2 foci localization throughout the *C. elegans* gonad (stained with DAPI in red and HA in yellow). Foci start to appear in early pachynema and increase in number throughout mid pachynema; in the late stages of pachynema, foci numbers are reduced. Scale bar: 10μm. (D) Mean numbers of RMIF-2::HA and HA::RMH-1 foci throughout pachynema: early pachynema, 5.6 (±5.8 SD) RMIF-2 foci (n = 221) and 5 (±5.1 SD) RMH-1 foci (n = 156 nuclei); mid pachynema, 9.0 (±4.6 SD) RMIF-2 foci (n = 184) and 9.4 (±4.7 SD) RMH-1 foci (n = 106); and late pachynema, 4.3 (±2.2 SD) RMIF-2 foci (n = 118) and 5.9 (±2.3 SD) RMH-1 foci (n = 63); three gonads per genotype. Significant differences were determined using a Student T-test: ns = not significant ($p > 0.05$); *** $p < 0.005$. Data are the mean and standard deviation (error bars). (E) Representative images of *C. elegans* mid/late pachynema nuclei stained with DAPI (magenta), HA (yellow) and GFP (cyan). RMH-1 and RMIF-2 foci co-localize in mid–late pachynema nuclei. Scale bar: 10μm.

**Table 2. Offspring analysis including embryonic lethality, brood size and segregation of male progeny in the genotypes used in this study.**

| Genotype | Embryonic lethality (%, mean ± SD) | Brood size (mean ± SD) | Males (%, mean ± SD) |
|---|---|---|---|
| WT | 1 ± 1 | 217 ± 35 | 0 |
| *rmif-2(jf113)* | 40 ± 4 | 129 ± 50 | 10 ± 3.8 |
| *rmh-1(jf54)* | 68 ± 9 | 130 ± 26 | 14 ± 5.6 |
| *rmif-2(jf113); rmh-1(jf54)* | 56 ± 37 | 72 ± 83 | 5 ± 3 |
| *rmif-2::ha* | 0.4 ± 0.5 | 210 ± 25 | 0 |
| *rmif-2::3xflag* | 0.3 ± 0.2 | 266 ± 26 | 0 |
| *him-6(ok412)* | 41 ± 5.6 | 202 ± 22 | 6 ± 2.6 |
| *rmif-2(jf113) him-6(ok412)* | 99.2 ± 1.6 | 15 ± 14 | 0 |
| *top-3::ollas* | 0.5 ± 0.4 | 269 ± 50 | 0 |
| *ha::degron::rmh-1* | 0.5 ± 0.5 | 210 ± 49 | 0 |

The CRISPR-Cas9 *rmif-2* deletion allele revealed a role in meiotic segregation. Embryonic lethality, reduced brood size, and a high incidence of males in the progeny suggest a defect in meiotic chromosome segregation. Counts are derived from the following numbers of hermaphrodites: WT, 10; *rmif-2*, 10; *rmh-1*, 10; *rmif-2;rmh-1*, 10; *rmif-2::ha*, 10; *rmif-2::3xflag*, 9; *him-6*, 10; *rmif-2 him-6*, 10; *top-3::ollas*, 14; and *ha::degron::rmh-1*, 6. A Mann-Whitney test for statistical differences in lethality was performed: WT vs *rmif-2* [****] (p<0.0001); WT vs *rmh-1* [***] (p<0.001); WT vs *rmif-2;rmh-1* [*] (p = 0.0134); WT vs *rmif-2::ha* (ns) (p = 0.3695); WT vs *rmif-2::3xflag* [*] (p = 0.0155); WT vs *him-6* [****] (p<0.0001); WT vs *rmif-2 him-6* [****] (p<0.0001); WT vs *top-3::ollas* ns (p = 0.5135); WT vs *ha::degron::rmh-1* ns (p = 0.6762); *rmif-2* vs *rmh-1* [*] (p = 0.0115); *rmif-2* vs *rmh-1;rmif-2* [*] (p = 0.0338); *rmh-1* vs *rmh-1;rmif-2* ns (p = 0.1014); *him-6* vs *him-6 rmif-2* [****] (p<0.0001); *rmif-2* vs *him-6 rmif-2* [****] (p<0.0001). WT–wild type.

resulting *rmif-2(jf113)* mutant exhibited several phenotypes characteristic of defective meiotic chromosome segregation. These included increased embryonic lethality (40% (± 4 SD)) and a high incidence of males (10% (± 3.8 SD)) among the progeny compared with controls (Table 2), both of which indicate chromosome mis-segregation in *C. elegans* [31]. Furthermore, the brood size was significantly reduced in *rmif-2(jf113)* mutants (129 (± 50 SD) versus 217 (± 35 SD) in the WT). Therefore, *rmif-2(jf113)* phenotypes are characteristic of a defect in meiotic recombination.

The number of DAPI-stained chromosome bodies in diakinesis nuclei can serve as a readout for meiotic prophase I events [32]. A WT *C. elegans* diakinesis nucleus contains six bivalents (chromosome pairs connected by chiasmata; Fig 2A and 2B). In contrast, diakinesis nuclei in the *rmif-2* mutant contained an increased number of DAPI-stained bodies, which likely represent a mix of univalents and bivalents (mean 6.9 (± 1 SD) bodies; Fig 2A and 2B). In addition, the *rmif-2* diakinesis phenotype was dependent on meiotic DSBs generated by the SPO-11 topoisomerase. In diakinesis, *rmif-2 spo-11* double mutants contained an average of 12 (± 0.2 SD) DAPI-stained bodies, indicating that the *rmif-2* diakinesis phenotype arose only after the induction of meiotic DSBs (Fig 2A and 2B).

To examine the role of *rmif-2* in meiotic DNA repair, we analyzed the formation and processing of meiotic recombination intermediates by monitoring the dynamic appearance and disappearance of the RAD-51 recombinase [33,34]. For this, we divided the gonads into seven equal zones and quantified the RAD-51 foci per nucleus in each zone (Fig 2C). In the WT, RAD-51 foci begin to accumulate in the transition zone and peak in mid pachynema; in late pachynema, the foci disappear as a consequence of successful repair. In *rmif-2*, RAD-51 foci appeared with similar dynamics as in the WT. However, greater numbers of foci accumulated and persisted throughout early/mid pachynema (Fig 2C) with a significantly different increase in zones 3–5; by late pachynema most of the foci had disappeared, and at diplonema no RAD-51 signal remained. Overall, DSB repair in *rmif-2* was delayed but accomplished eventually. We also found that aberrant RAD-51 accumulation was SPO-11 dependent (as shown in the *rmif-2 spo-11* double mutant; Fig 2C). There was no significant difference between the RAD-

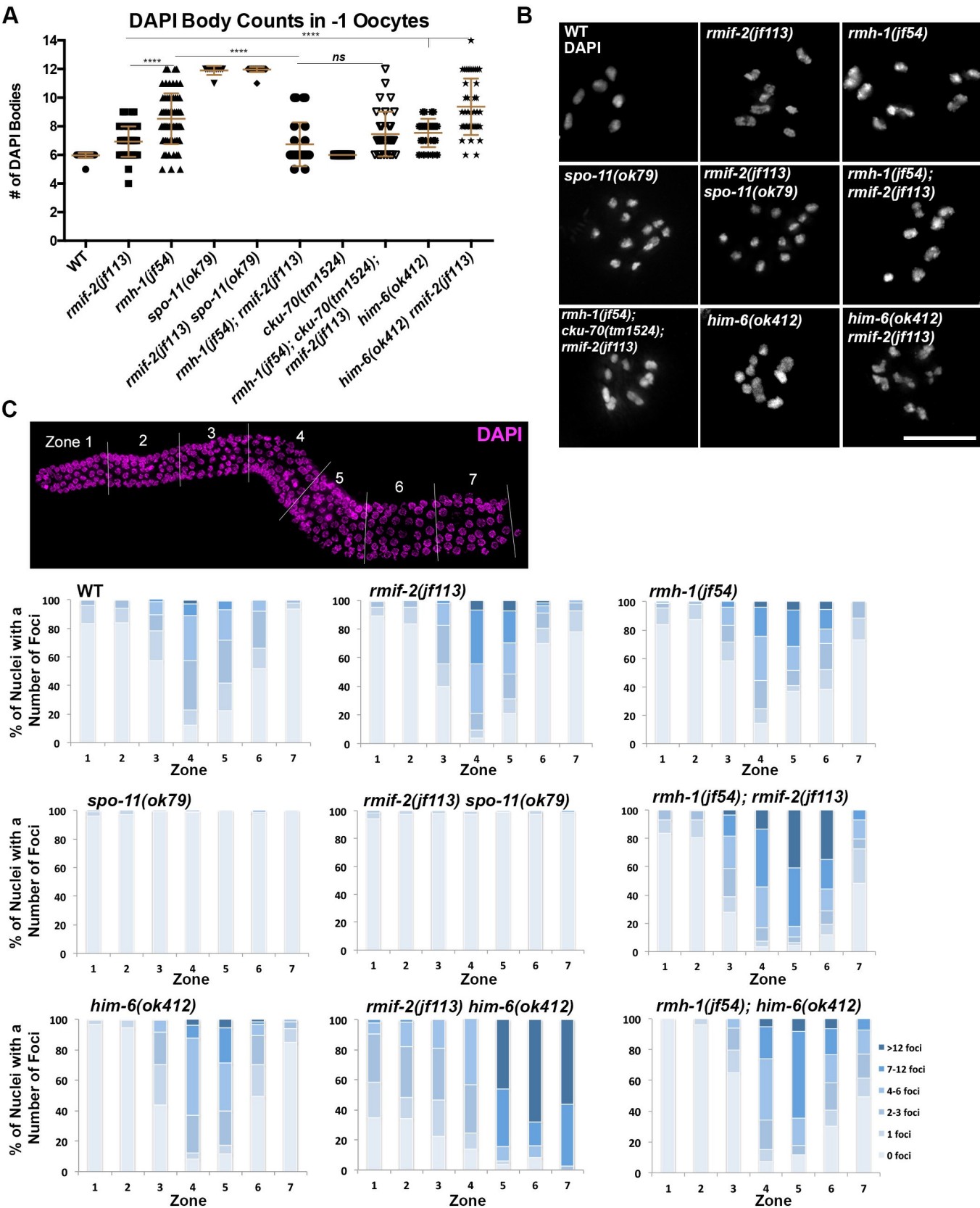

**Fig 2. RMIF-2 is required for robust chiasma formation and chromosome segregation in meiosis.** (A) Quantification of DAPI-stained bodies in -1 diakinesis oocytes in the WT (number of nuclei, n = 32), *rmif-2(jf113)* (n = 41), *rmh-1(jf54)* (n = 74), *spo-11(ok79)* (n = 12), *rmif-2(jf113) spo-11(ok79)* (n = 32), *rmh-1 (jf54); rmif-2(jf113)* (n = 27), *cku-70(tm1524)* (n = 19), *rmh-1(jf54); cku-70(tm1524); rmif-2(jf113)* (n = 33), *him-6(ok412)* (n = 26), and *rmif-2(jf113) him-6 (ok412)* (n = 41) mutants. Data are the mean and standard deviation (error bars). Significant differences were determined using a Student T-test: **** $p < 0.0001$. (B) Representative images of chromosomes in a diakinesis nucleus for each genotype stained with DAPI. Scale bar: 10μm. (C) Quantification of RAD-51 profiles throughout meiotic prophase I (upper panel). *C. elegans* gonads were divided into seven equal zones. RAD-51 foci were counted in each nucleus of each zone; three representative gonads per genotype. Graphs show the percentage of nuclei with different numbers of foci per germline zone. Raw data and statistical analysis of RAD-51 profiles between different zones and genotypes via Fisher's exact test are presented in S2 File. Scale bar: 10μm.

51 foci counts in the *spo-11* and *rmif-2 spo-11* mutants. This indicates a specific defect in the processing of meiotic recombination intermediates in *rmif-2*.

Concomitant with recombination, chromosome pairing is initiated soon after meiotic entry [32]. We used the X chromosome-binding protein HIM-8 as a specific tool to analyze X chromosome pairing [35]. In the *rmif-2* mutant, X chromosome pairing reached WT levels in pachynema (S2A Fig), although with slower kinetics up to early pachynema. We assume that this is a consequence of the slightly extended mitotic zone in the *rmif-2* mutant (mean 23.4 (± 3.0 SD) cell rows versus 20.4 (± 2.6 SD) in the WT; S2B Fig). Progressive co-localization of the HTP-3 chromosome axis marker and SYP-1 synapsis marker serves as a read-out for synaptonemal complex formation [36]. Using this read-out, no major defects in synapsis were detected in the *rmif-2* mutant (S2C Fig).

Taken together, the phenotypic data reveal that *rmif-2* mutants display aberrant recombination and univalent diakinesis chromosomes, which goes in hand with an increased embryonic lethality and X chromosome non-disjunction. All of these phenotypes have also been reported for *rmh-1* mutants, although to differing degrees. In *rmh-1* mutants, both the number of univalents and the degree of embryonic lethality are higher (Fig 2A and 2B, and Table 2) and [19].

## RMIF-2 is required to concentrate the other BTR complex proteins into recombination foci

In tissue-cultured cells, RMI2 is required to localize the Bloom helicase into recombination foci [27], and RMI2 interaction with the other BTR complex proteins is mediated by the OB-fold domain [28]. We therefore wanted to examine whether the correct localization of all BTR complex proteins depends on RMI2. We first analyzed GFP::RMH-1 localization in the *rmif-2* mutant. In the *gfp::rmh-1*, faint RMH-1 foci are seen throughout early pachynema, and become brighter from mid to late pachynema (Fig 3A) [19]. GFP::RMH-1 was not detectable in the *rmif-2* mutant: only faint cytoplasmic foci were occasionally seen (Fig 3A), suggesting that RMIF-2 is essential for RMH-1 localization into discrete chromatin-associated foci. To address whether RMIF-2 stabilizes the RMH-1 protein, we isolated germline-enriched nuclei, as described in [29], followed by subcellular fractionation and western blot analysis (Fig 3B and 3C). RMH-1 was enriched in both the soluble and insoluble nuclear fractions in the *ha:: rmh-1* (Fig 3B and 3C). In the absence of RMIF-2, the soluble fraction contained less RMH-1 and the insoluble fraction was below levels of detection (Fig 3B and 3C). Next, we addressed the reciprocal question of whether RMH-1 stabilizes the RMIF-2 protein. In *rmh-1* mutants, RMIF-2 foci were undetectable (Fig 3D). Western blot analysis of whole-cell extracts showed that RMH-1 stabilizes RMIF-2 (Fig 3E): the quantity of RMIF-2 protein was three times lower in the *rmh-1; rmif-2::ha* mutant than in the *rmif-2::ha* (Fig 3F). These data indicate a reciprocal requirement for RMH-1 and RMIF-2 for their recruitment into recombination foci and for protein stability.

To investigate further the dependency of the BTR complex members we examined the localization of HIM-6 helicase in the *rmif-2* mutant. In *him-6::ha*, HIM-6 localizes into bright

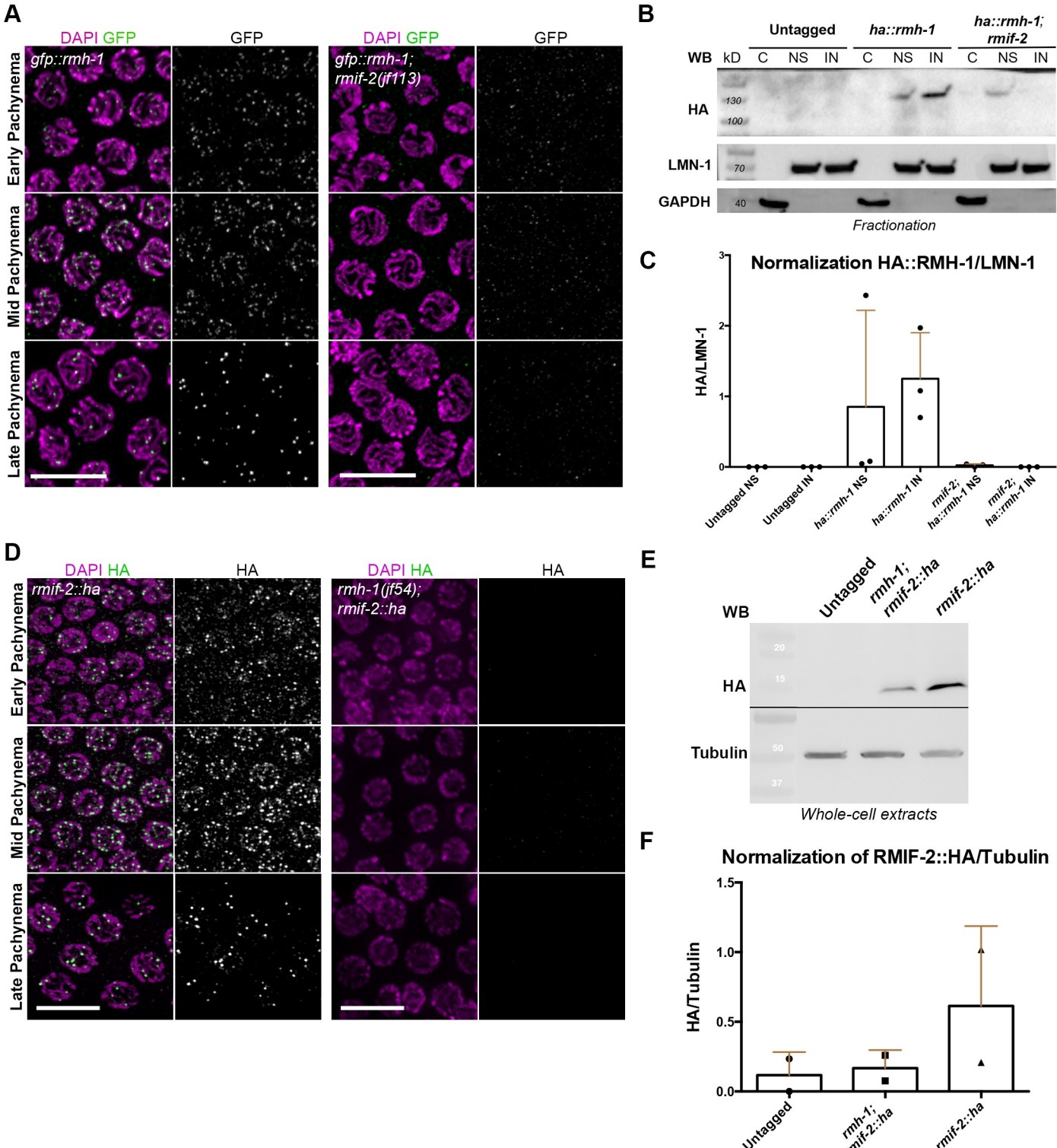

**Fig 3. Chromatin loading and abundance of RMIF-2 and RMH-1 proteins are mutually dependent.** (A) Representative images of *gfp::rmh-1* and *gfp::rmh-1; rmif-2(jf113)* pachytene nuclei stained with DAPI (magenta) and GFP (green). GFP::RMH-1 localization to nuclear foci starts in early pachynema, peaks in mid pachynema, and becomes concentrated in six foci in late pachynema. In the *rmif-2* mutant background, RMH-1 fails to localize into foci throughout pachynema, except for a very few cytoplasmic foci. Scale bar: 10μm. (B) A protein fractionation shows specific HA::RMH-1 enrichment in the nucleus, which is reduced in the *rmif-2* mutant. Equal amounts of protein were loaded for each fraction. C = cytosolic fraction, NS = soluble nuclear fraction, IN = insoluble nuclear fraction. LMN-1 was the loading control for nuclear fractions; GAPDH was the loading control for the cytosolic fraction. (C) Western blot normalization and quantification of

nuclear fractionations from untagged WT, *ha::rmh-1*, and *ha::rmh-1; rmif-2* samples. Three biological replicates were used for each sample. (D) Representative images of *rmif-2::ha* and *rmh-1(jf54); rmif-2::ha* pachytene nuclei stained with DAPI (magenta) and HA (green). RMIF-2 localization to nuclear foci throughout meiotic prophase starts in early pachytene nuclei, peaks in mid pachynema, and decreases in late pachynema. In the *rmh-1* mutant background, RMIF-2 fails to localize to nuclear foci throughout pachynema. Scale bar: 10μm. (E) Western blot analysis of RMIF-2::HA in whole-cell extracts in WT (untagged) and the *rmh-1* mutant background. WT worms were used to test the antibody specificity. The predicted size of RMIF-2::HA is 15kD. Tubulin was the loading control. (F) Western blot normalization and quantification of Untagged WT, *rmh-1; rmif-2::ha* and *rmif-2::ha* mutants. Two biological replicates were used.

foci throughout pachynema, but these foci appeared smaller and fainter in the absence of RMIF-2 (Fig 4A) [19]. Previously it was shown that in the *rmh-1* mutant, HIM-6 foci were present throughout pachynema but seemed smaller and fainter, suggesting that RMH-1 is required to stabilize and enrich HIM-6 into foci [19]. Taken together these data suggest that the initial recruitment of HIM-6 is independent of both RMIF-2 and RMH-1 [19], but both of these factors seem to be necessary for HIM-6 accumulation in recombination foci.

To visualize TOP-3, we inserted an internal OLLAS tag into the protein coding sequence (see Materials and Methods). The tagged line was functional since it had a hatching rate of 99.5% ± 0.4 SD (Table 2) compared with 0% (i.e. 100% embryonic lethality) in the null mutant [37]. TOP-3 also localized to foci throughout pachynema, with similar dynamics and foci numbers to RMH-1 and RMIF-2 (Fig 4B–4D). The number of TOP-3 foci peaked in mid pachynema, while in late pachynema TOP-3 appeared to be present on the putative six CO sites. TOP-3 failed to localize properly in *rmif-2*, with the TOP-3 signal detectable as greatly reduced, sporadic foci (Fig 4B–4D), consistent with a requirement for RMIF-2 in TOP-3 complex stabilization. Taken together, these data suggest that constituents of the BTR complex require RMIF-2 for their proper localization into foci throughout pachynema. Although the initial recruitment of HIM-6 appears to be independent of RMIF-2, RMIF-2 seems to enhance the enrichment of HIM-6 into foci. These findings are consistent with a model in which (1) RMIF-2 stabilizes the BTR complex and (2) RMH-1 and RMIF-2 recruitment into discrete chromatin-associated foci is essential for the mutual stabilization of both proteins.

## Localization of COSA-1 and MSH-5 pro-CO factors in the *rmif-2* and *rmh-1* mutants

The establishment of inter-homolog COs requires several proteins, including the *C. elegans* COSA-1/CNTD1 cyclin [38,39] and the MutSγ complex MSH-4/MSH-5 [40–43]. We previously reported that RMH-1 fails to stably localize into recombination foci (composed of both CO and NCO intermediates) in the *msh-5* and *cosa-1* mutants and, consequently, that the six bright foci representing CO intermediates were missing in late pachynema [19]. COSA-1 foci were also markedly reduced in *rmh-1* mutants, suggesting a decreased efficiency in CO designation. Therefore, we used an OLLAS::COSA-1 fusion protein [44] to compare COSA-1 dynamics in the *rmif-2* and *rmh-1* mutants (Fig 5A–5C). COSA-1 foci started to appear in zone 4 (as defined in Fig 2C), corresponding to early–mid pachynema (range 0–13 foci per nucleus; mean 6.2 (± 2.4 SD) in zone 5 and 5.5 (± 1.2 SD) in zone 6). In late pachynema (zone 7), an average of 6 (± 0.2 SD) very bright COSA-1 foci were observed in the *ollas::cosa-1* (Fig 5A and 5D). Interestingly, the foci profiles differed in *rmh-1; ollas::cosa-1* and *ollas::cosa-1; rmif-2*, (Fig 5B–5D). In both *rmh-1* and *rmif-2* mutants, COSA-1 foci accumulation was significantly delayed (starting in zone 6 and 5, respectively) and significantly fewer foci were observed (zone 4: *ollas::cosa-1* (2.4 ± 3.9 SD), *ollas::cosa-1; rmif-2* (0 ± 0 SD), *rmh-1; ollas::cosa-1* (0 ± 0 SD); zone 5: *ollas::cosa-1* (6.2 ± 2.4 SD), *ollas::cosa-1; rmif-2* (1 ± 1.7 SD), *rmh-1; ollas:: cosa-1* (0 ± 0 SD); zone 6: *ollas::cosa-1* (5.5 ± 1.2 SD), *ollas::cosa-1; rmif-2* (4.4 ± 1.8 SD), *rmh-1; ollas::cosa-1* (1.7 ± 1.4 SD); zone 7: *ollas::cosa-1* (5.9 ± 0.2 SD), *ollas::cosa-1; rmif-2* (4.9 ± 1.5 SD), *rmh-1; ollas::cosa-1* (3 ± 1.4 SD); (Fig 5D)).

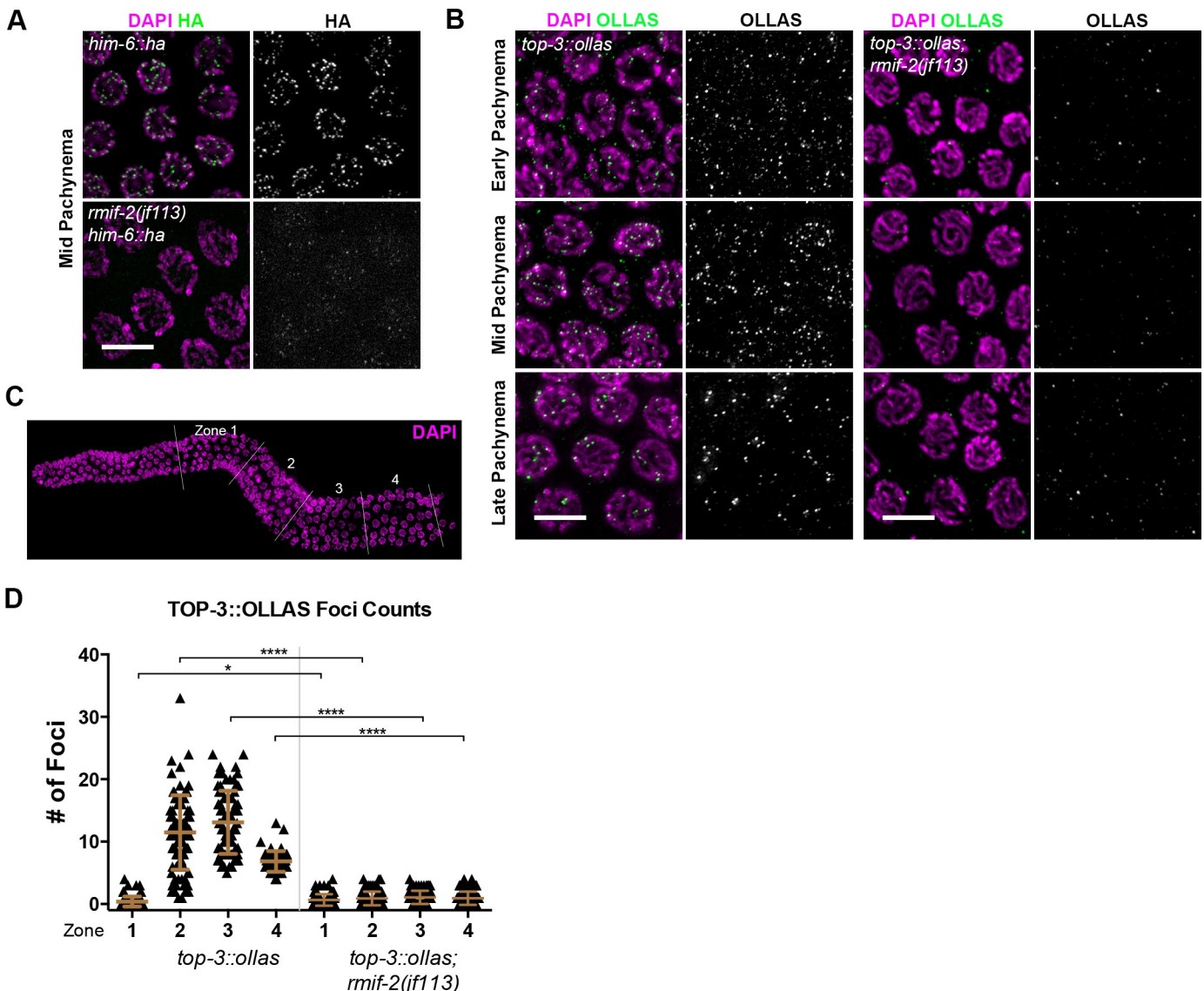

**Fig 4. Localization of HIM-6 and TOP-3 in the *rmif-2* mutant.** (A) Representative *him-6::ha* and *rmif-2(jf113) him-6::ha* nuclei in mid pachynema stained with DAPI (in magenta) and HA (in green). HIM-6 localizes to bright foci throughout pachynema in *him-6::ha*. In the *rmif-2* mutant, HIM-6 is detected in small, faint foci throughout pachynema. Scale bar: 5 µm. (B) Representative images of nuclei throughout pachynema stained with DAPI (in magenta) and OLLAS (in green). TOP-3::OLLAS localizes to distinct foci throughout early, mid, and late pachynema. In the *rmif-2* mutant, TOP-3 fails to localize properly, and only a few cytoplasmic and nuclear foci can be observed. Scale bars: 5 µm. (C) For the quantification of TOP-3::OLLAS foci three gonads per genotype were each divided into four equal zones from the transition zone (beginning of meiosis) until late pachynema. (D) Quantification of TOP-3 foci in *top-3::ollas* and *top-3::ollas; rmif-2* backgrounds, throughout the *C. elegans* gonad. The mean number of TOP-3 foci in each zone was WT: zone 1: 0.4 (±0.8 SD), n = 97 nuclei; zone 2: 11.5 (±6 SD), n = 89; zone 3: 13.1 (±5 SD), n = 79; and zone 4: 6.8 (±1.6 SD), n = 57; *rmif-2*: zone 1: 0.6 (±0.9 SD), n = 103 nuclei; zone 2: 0.9 (±1.0 SD), n = 109; zone 3: 1.0 (±1.0 SD), n = 94; zone 4: 0.7 (±0.98 SD), n = 57.

Like COSA-1, the MSH-4/MSH-5 heterodimer is essential to install the COs on the six homolog pairs in *C. elegans* [40,41] to support the maturation of CO-designated sites into COs. MSH-5 localizes to numerous foci throughout pachynema, marking both the different stages of maturing COs and also NCO intermediates [45,46]. In the *gfp::msh-5* tagged line, we detected MSH-5 [44] in foci in zones 3–6 (range 0–19 foci/nucleus) and in zone 7 (mean 5.1 (± 2.5 SD) foci/nucleus) (Fig 6A and 6B). These were previously reported to co-localize with

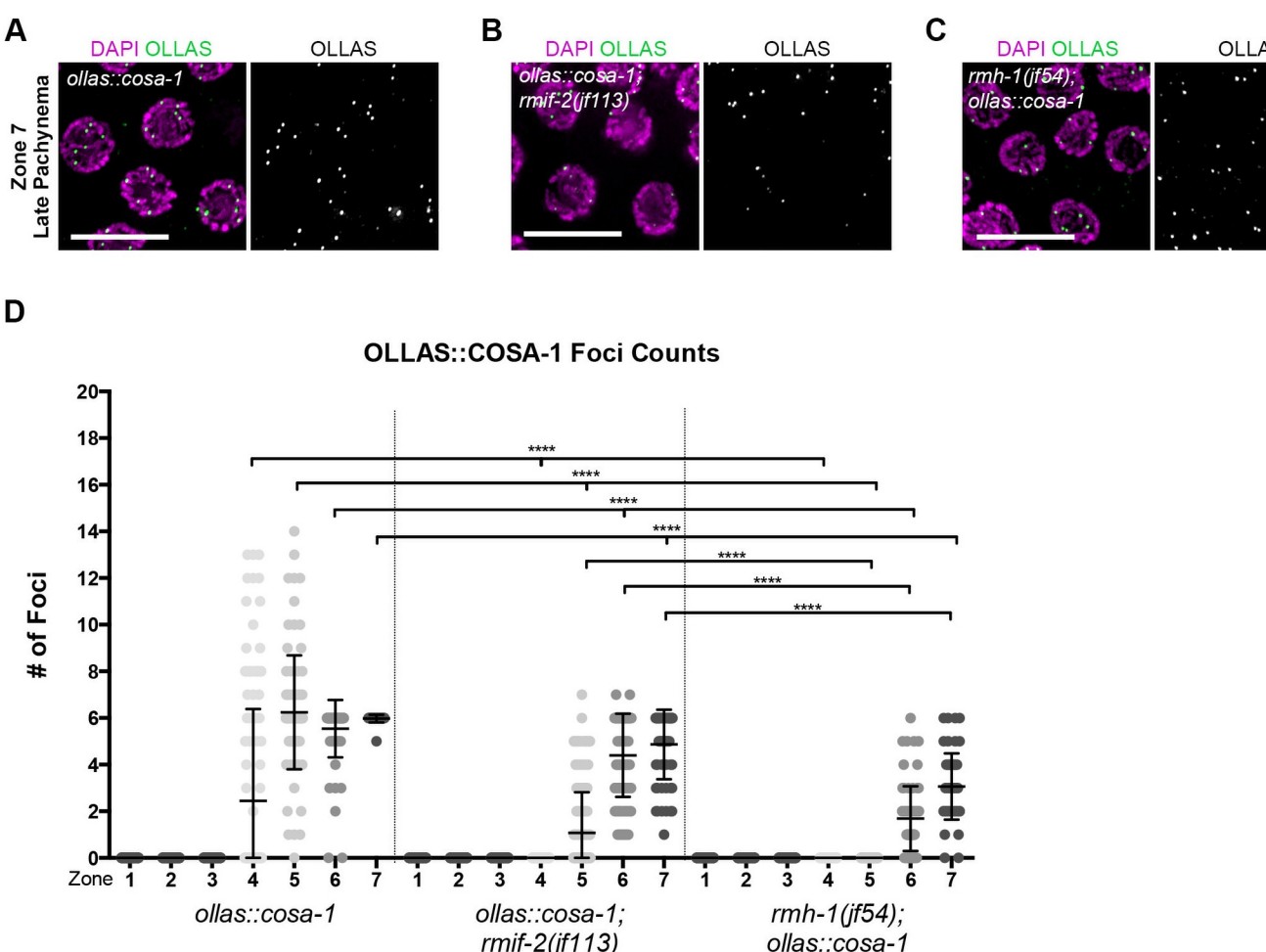

**Fig 5. Analysis of the recombination marker OLLAS::COSA-1 in the *rmif-2* and *rmh-1* mutants.** (A–C) Representative images (top) stained for DAPI (in magenta) and OLLAS (in green) in late pachynema (zone 7; defined in Fig 2) and quantification of OLLAS::COSA-1 nuclear foci (bottom) in the *ollas::cosa-1*, in *ollas::cosa-1; rmif-2(jf113)* and *rmh-1(jf54); ollas::cosa-1* mutants. Scale bars, 10 μm. (D) Gonads were divided into seven equal zones from the mitotic tip to late pachynema (n = 3 gonads per genotype). Significant differences in foci distribution were determined using a Mann-Whitney test: zone 4: *ollas::cosa-1* vs *ollas::cosa-1; rmif-2* **** (p<0.0001); *ollas::cosa-1* vs *rmh-1; ollas::cosa-1* **** (p<0.0001); *ollas::cosa-1; rmif-2* vs *rmh-1; ollas::cosa-1* ns (p>0.9999). Zone 5: *ollas::cosa-1* vs *ollas::cosa-1; rmif-2* **** (p<0.0001); *ollas::cosa-1* vs *rmh-1; ollas::cosa-1* **** (p<0.0001); *ollas::cosa-1; rmif-2* vs *rmh-1; ollas::cosa-1* **** (p<0.0001). Zone 6: *ollas::cosa-1* vs *ollas::cosa-1; rmif-2* **** (p<0.0001); *ollas::cosa-1* vs *rmh-1; ollas::cosa-1* **** (p<0.0001); *ollas::cosa-1; rmif-2* vs *rmh-1; ollas::cosa-1* **** (p<0.0001). Zone 7: *ollas::cosa-1* vs *ollas::cosa-1; rmif-2* **** (p<0.0001); *ollas::cosa-1* vs *rmh-1; ollas::cosa-1* **** (p<0.0001); *ollas::cosa-1; rmif-2* vs *rmh-1; ollas::cosa-1* **** (p<0.0001).

the CO markers COSA-1, ZHP-3, and RMH-1 [19]. In the *rmif-2; gfp*::*msh-5* mutant, MSH-5 displayed a dynamic foci profile as in the *gfp*::*msh-5* (range 0–18 foci/nucleus; mean: zone 4: 1 (± 2.5 SD), zone 5: 5.9 (± 5 SD), zone 6: 5 (± 2.0 SD), zone 7: 4.8 (± 1.6 SD) foci/nucleus; (Fig 6A and 6B)). Clearly, MSH-5 accumulates into distinct foci in *rmif-2*; nevertheless, their appearance was delayed and the numbers were significantly reduced in comparison to the *gfp*::*msh-5*. Furthermore, when we co-stained the nuclei with the CO marker ZHP-3, we observed the co-localization of MSH-5 and ZHP-3, suggesting that MSH-5 foci mark CO sites (Fig 6C). Co-staining of ZHP-3 and GFP was not efficient enough to allow quantification of the signals; however, the staining pattern suggests that MSH-5 foci do represent CO sites. Therefore, we next quantified the degree of co-localization between the pro-CO markers COSA-1 and ZHP-3 in late pachynema in the *ollas::cosa-1* and *ollas::cosa-1; rmif-2* backgrounds. We found that

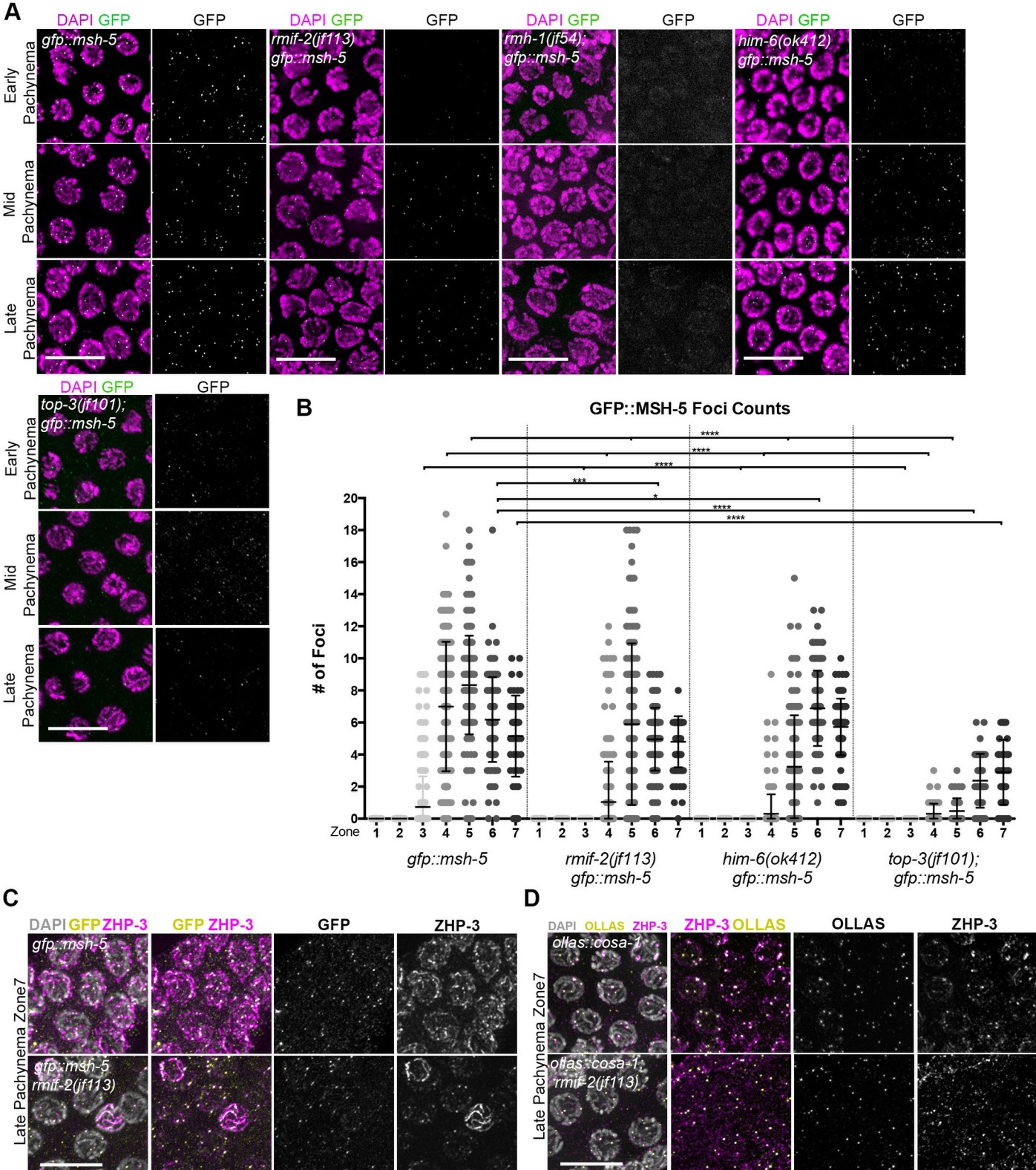

**Fig 6. GFP::MSH-5 localization in BTR complex mutants.** (A) Representative images of GFP::MSH-5 nuclear foci (in green) and DAPI (in magenta) in early, mid, and late pachynema in the *gfp::msh-5* and *rmif-2(jf113) gfp::msh-5*, *rmh-1(jf54); gfp::msh-5*, *him-6(ok412) gfp::msh-5* and *top-3(jf101); gfp::msh-5* mutants. Scale bars, 10 μm. (B) Quantification of GFP::MSH-5 nuclear foci in *gfp::msh-5*, *rmif-2(jf113) gfp::msh-5*, *him-6(ok412) gfp::msh-5* and *top-3(jf101); gfp::msh-5* mutants. Gonads

were divided into seven equal zones from the mitotic tip to late pachynema (n = 3 gonads per genotype). Significant differences in foci distribution were determined using a Mann-Whitney test. Zone 3: *gfp::msh-5* vs *rmif-2 gfp::msh-5* **** (p<0.0001); *gfp::msh-5* vs *him-6 gfp::msh-5* **** (p<0.0001); *gfp::msh-5* vs *top-3; gfp::msh-5* **** (p<0.0001). Zone 4: *gfp::msh-5* vs *rmif-2 gfp::msh-5* **** (p<0.0001); *gfp::msh-5* vs *him-6 gfp::msh-5* **** (p<0.0001); *gfp::msh-5* vs *top-3; gfp::msh-5* **** (p<0.0001). Zone 5: *gfp::msh-5* vs *rmif-2 gfp::msh-5* **** (p<0.0001); *gfp::msh-5* vs *him-6 gfp::msh-5* **** (p<0.0001); *gfp::msh-5* vs *top-3; gfp::msh-5* **** (p<0.0001). Zone 6: *gfp::msh-5* vs *rmif-2 gfp::msh-5* *** (p = 0.0003); *gfp::msh-5* vs *him-6 gfp::msh-5* * (p = 0.0279); *gfp::msh-5* vs *top-3; gfp::msh-5* **** (p<0.0001). Zone 7: *gfp::msh-5* vs *rmif-2 gfp::msh-5* ns (p = 0.0501); *gfp::msh-5* vs *him-6 gfp::msh-5* ns (p = 0.3057); *gfp::msh-5* vs *top-3; gfp::msh-5* **** (p<0.0001). (C) Representative images of GFP::MSH-5 (yellow) and ZHP-3 (magenta) co-localization in late pachytene nuclei in *gfp::msh-5* and *rmif-2(jf113) gfp::msh-5* backgrounds. Scale bar, 10 µm. (D) Representative images of OLLAS (yellow) and ZHP-3 (magenta) co-localization in late pachytene nuclei in *ollas::cosa-1* and *ollas::cosa-1; rmif-2(jf113)* backgrounds. Scale bar, 10 µm.

94% of COSA-1 foci co-localized with ZHP-3 foci in the *rmif-2* mutant, compared with 98% in the WT (Fig 6D). In striking contrast, in the *rmh-1* mutant, MSH-5 failed to localize into detectable foci throughout pachynema (Fig 6A); instead, the protein appears as a hazy nuclear signal.

Interestingly, *rmh-1* was the only mutant in the BTR complex that failed to localize MSH-5 into clearly visible foci. In both the *him-6* and *top-3* mutants, GFP::MSH-5 could be detected in foci, albeit with a delayed appearance and in significantly reduced numbers. In *him-6* mutants, the MSH-5 signal can be seen throughout zones 4–7 (range 0–15 foci/nucleus), with an average of 5.7 (± 1.8 SD) foci per nucleus in zone 7 (Fig 6A and 6B). In the *top-3* mutant, the number of MSH-5 foci was much lower (range 0–6 foci/nucleus in zones 4–7) (Fig 6B). Although their appearance was delayed, foci were visible as distinct signals in zones 4–7 (mean 2.9 (± 2.0 SD) foci/nucleus in zone 7). Taken together, our data show that in all mutants of BTR complex proteins, CO designation is less efficient as depicted by the significantly delayed and reduced appearance of COSA-1 and MSH-5 foci, with the exception of the *rmh-1* mutant, where GFP::MSH-5 foci are completely absent.

Therefore, we next asked whether formation of the putative bivalent chromosomes observed in the diakinesis nuclei of *rmh-1* and *rmif-2* mutants (Fig 2A and 2B) depended on the canonical CO pathway [39,41]. Indeed, these chromosome structures were *cosa-1* dependent in the *rmif-2* mutant (S3A and S3B Fig). In contrast to the *rmif-2* single mutant, in the *cosa-1; rmif-2* double mutant diakinesis nuclei contained mostly univalents, as in *cosa-1* single mutants. The data indicate that the diakinesis joint chromosome structures observed in *rmif-2* are mostly formed via the class I CO pathway. As expected, RMIF-2 did not localize into recombination foci in the *cosa-1* mutant (S3C Fig). Instead, in mid pachynema RMIF-2 was observed in very few, faint foci, with most of the signal located outside the nucleus, and in late pachynema no nuclear foci were detectable.

Although MSH-5 foci were absent in *rmh-1* (where the MSH-5 signal forms a nuclear haze (Fig 6A)), we examined whether the putative bivalent chromosomes in *rmh-1* were still dependent on *msh-5*. Analysis of DAPI bodies in the *rmh-1; msh-5* double mutant (S3A and S3B Fig) revealed that many of the joint structures seen in diakinesis were indeed dependent on MSH-5. On average, 8.5 (±1.7 SD) DAPI-stained bodies were formed in *rmh-1*, 12 (± 0 SD) in *msh-5* and 10.8 (±1.4 SD) in *rmh-1; msh-5*. To further analyze whether the remaining physical attachments between univalent chromosomes in *rmh-1* mutants were dependent on non-homologous end joining (NHEJ) activity, we constructed the triple mutant *rmh-1; cku-70; msh-5*, (S3A and S3B Fig). CKU-70 is a key protein component of the NHEJ pathway [47]. Analysis of diakinesis chromosomes in the triple mutant revealed a significant increase in fragmentation, leading to on average 12 (± 0.5 SD) DAPI-stained bodies compared to the average of 6 (± 0 SD) in the *cku-70* single mutant and 10.8 (± 1.4 SD) in the double mutant *rmh-1; msh-5*, suggesting that some of the *rmh-1; msh-5* undefined structures (which seem larger than univalents) are dependent on the activity of the NHEJ pathway.

## RMIF-2 and RMH-1 may not always act interdependently in meiosis

To investigate whether RMIF-2 acts as a BTR complex stabilizer by binding to and or stabilizing RMH-1, we generated the *rmh-1; rmif-2* double mutant and compared its phenotype with those of the single mutants. First, we examined embryonic lethality and the number of males in the viable progeny. We observed that in the *rmh-1; rmif-2* double mutant, levels of embryonic lethality were similar to those in *rmh-1* mutants, 56% (± 37 SD) although with much higher variability between individual worms, with 5% (± 3 SD) of males in the progeny (Table 2; in comparison, embryonic lethality is 40% (± 4 SD) in *rmif-2* and 68% (± 9 SD) in *rmh-1* mutants). In the double mutant, the brood size was much smaller (mean 72 (± 83 SD)) than in the single mutants (Table 2). We next examined DAPI-stained diakinesis chromosomes in *rmh-1; rmif-2* worms. There were an average of 6.7 (± 1.5 SD) DAPI bodies per nucleus, and most nuclei contained six aberrant bodies that differed markedly from the well-shaped bivalents in the WT (Fig 2A and 2B). We saw chromatin clumps together with DAPI bodies of differing sizes that resembled DNA fragments and/or univalents (Fig 2B). We wanted to examine whether these DAPI-stained bodies were the results of NHEJ activity and thus, constructed the triple mutant *rmh-1; cku-70; rmif-2* (Fig 2A and 2B). Analysis of diakinesis nuclei in this mutant background revealed no difference in the number of DAPI bodies (7.45 (± 1.6 SD)) when compared to *rmh-1; rmif-2* (6.7 (± 1.5 SD)). Therefore NHEJ does not cause the chromosome abnormalities observed at diakinesis stage. We also analyzed DSB induction and repair in the double mutant *rmh-1; rmif-2*. Assessment of RAD-51 loading/unloading on chromosomes revealed a striking accumulation of RAD-51 foci, which persisted throughout pachynema (statistically different from the single mutants; Fig 2C). Exacerbation of the meiotic phenotypes in the *rmh-1; rmif-2* double mutant suggests that either RMIF-2 or RMH-1 may have an additional function outside of the mutually dependent formation of recombination foci and/or that RMIF-*2* might not merely function as a RMH-1 stabilizer within the BTR complex. It could be that RMIF-2 has a less prominent role during meiotic DSB repair.

We next generated the *rmif-2 him-6* double mutant to test whether RMIF-2 was needed to support the activity of the HIM-6 helicase. In these worms, DAPI-body counts showed an increase to an average of 9.4 (± 2.0 SD) in comparison to the single mutants *rmif-2*, 6.9 (± 1.0 SD) and *him-6*, 7.5 (± 0.9 SD) (Fig 2A and 2B). In contrast, in the *rmh-1; him-6* diakinesis DAPI-bodies were not statistically different in comparison to the *him-6* single mutant (7.8 (± 1 SD) vs 7.3 (± 1.1 SD)) [19]. Moreover, in *rmif-2 him-6* embryonic lethality strikingly increased to 99.2% (± 1.6 SD) from 41% (± 5.6 SD) in the *him-6* single mutant (Table 2), while in the *rmh-1; him-6* it was 68.22% [19]. The quantification of RAD-51 in the double mutant *rmif-2 him-6* revealed accumulation of meiotic and mitotic recombination intermediates. RAD-51 was loaded earlier than in the WT and single mutants, from the mitotic zone (zone 1) onwards, with high and persisting numbers of unresolved breaks, which were not repaired by late pachynema (zone 7) (Fig 2C). Taken together, RAD-51 quantification revealed high levels of genomic instability in the *rmif-2 him-6* animals arising from mitotic and meiotic defects. The severe phenotype of *rmif-2 him-6* double mutants suggests that RMIF-2 and HIM-6 might act in independent parallel pathways. Analysis of the accumulation of recombination intermediates in the *rmh-1; him-6* revealed differences in comparison to the *rmif-2 him-6* double mutant. Here, RAD-51 was also observed in significantly higher numbers throughout pachynema, with accumulation of unrepaired breaks until late pachynema (Fig 2C), however no significant amount of RAD-51 signals was quantified in the mitotic zone (zones 1–2).

## RMIF-2 and RMH-1 influence the recombination landscape

Based on observed differences in the localization dynamics of key recombination factors in *rmif-2* and *rmh-1* mutants, we performed a recombination analysis to compare the

recombination rate and CO number and position in the mutants with those of the WT. For this, we introduced homozygous mutations into a *C. elegans* hybrid strain derived from the Bristol and Hawaii isolates. Single nucleotide polymorphisms (SNPs) on chromosomes IV and V were used for assessing COs [19,48] (Fig 7A). PCR-based analysis of SNPs on chromosomes IV and V was conducted on F2 worms (described in Materials and Methods). As previously observed for *rmh-1*, we did not find significant differences between the total frequencies of COs in *rmif-2* compared to WT (on both chromosomes IV and V), and in *rmh-1* on chromosome V (Fig 7). We observed a significant shift of COs towards the central regions of both chromosome IV and V, where CO are usually not favored in the WT (Fig 7B–7D). In addition, an increase in double and triple COs was observed in the *rmif-2* in contrast to *rmh-1* that displayed only few extra COs. For chromosome IV two double COs were recorded for *rmif-2* (n = 364) and one for the WT (n = 281). On chromosome V nine double COs and two triple COs were observed in *rmif-2* (n = 362); in contrast, only one double CO was seen in the WT (n = 269) (Fig 7D). In summary, fewer extra COs were present in the *rmh-1* mutant than in the *rmif-2* mutant. Taken together, the recombination assays revealed that both RMIF-2 and RMH-1 play a role in correctly positioning COs to chromosome arms (away from the chromosome center), with RMIF-*2* having a more pronounced role in suppressing the formation of double and triple COs [49].

In *C. elegans*, HIM-6 (BLM) helicase is involved in rejecting strand invasion into heterologous sequences [12], and lack of this activity leads to genome rearrangements. We used visible phenotypic markers to assess the extent of heterologous recombination within the *mIn1* inversion on chromosome II [12]: one copy of chromosome II is marked with the semi-dominant *dpy-25* mutation and the second copy contains the *mIn1* inversion, which is marked with the recessive *rol-1* mutation and has a semi-dominant insertion of a GFP-expressing transgene (Fig 8A). In WT worms, heterologous recombination is rare (Fig 8B), as previously reported [12], and we did not detect a single event (n = 2029). In contrast, in absence of RMIF-2, there was a significant increase in progeny displaying heterologous recombination events (2.3%, n = 2018) (Fig 8B and 8C). Surprisingly, the level was roughly three times higher in the *rmh-1* mutants, at 7.24% (n = 1090), consistent with the level seen in *him-6* mutants (6.6%), as reported in [12]. Taken together, these data show that both *rmif-2* and *rmh-1* are required to prevent heterologous recombination; however, the more pronounced defect in *rmh-1* mutants compared with *rmif-2* worms suggests that RMIF-2 might have a less prominent role in rejecting heterologous strand invasion.

## Discussion

In this study, we present a *C. elegans* functional homolog of the BTR complex stabilizer RMI2. The newly identified RMIF-2 protein contains a characteristic OB-fold domain and is also found in RMH-1-containing protein complexes. Our analysis in meiosis prophase I showed that the recombination foci that mark both CO and NCO recombination sites during pachynema contain HIM-6, TOP-3, RMH-1, and RMIF-2. In late pachynema, these foci decorate the obligate COs on the six *C. elegans* bivalents. The concentration of most BTR complex proteins into these foci depended on the presence of RMIF-2. The exception was the HIM-6 helicase, which was recruited independently, although its enrichment at recombination sites was RMIF-2 dependent. Furthermore, similar to reported observations in mitotic cells [27,28], we showed that RMIF-2-dependent concentration of RMH-1 into recombination foci strongly influenced the RMH-1 protein stability.

The *rmh-1* and *rmif-2* single mutants shared phenotypical features, indicating a role in both CO and NCO formation and in the suppression of heterologous recombination. Phenotypes of both mutants included embryonic lethality, segregation of males (through non-disjunction

**A**

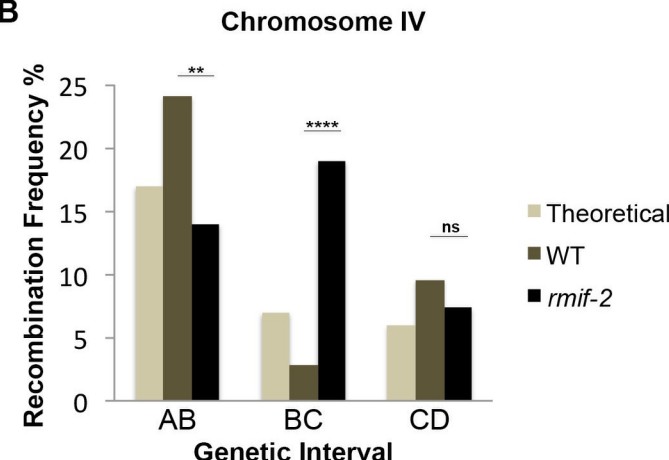
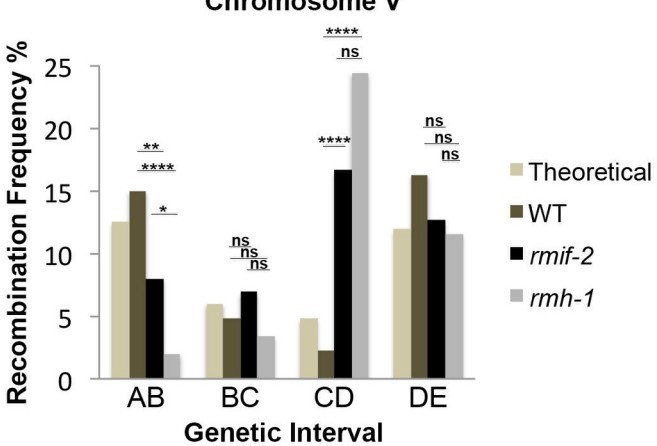

**B**

Chromosome IV and Chromosome V recombination frequency bar charts.

**C**

| Chr IV | AB | BC | CD |
|---|---|---|---|
| WT | 66.02 (68) | 7.8 (8) | 26.2 (27) |
| *rmif-2* | 34.3 (50) | 47.2 (69) | 18.5 (27) |

| Chr V | AB | BC | CD | DE |
|---|---|---|---|---|
| WT | 38.2 (39) | 12.7 (13) | 5.9 (6) | 43.13 (44) |
| *rmif-2* | 18.1 (29) | 15.6 (25) | 37.5 (60) | 28.75 (46) |
| *rmh-1* | 7.7 (8) | 13.5 (14) | 52.8 (56) | 25 (26) |

**D**

| Chr IV | SCO | DCO | TCO |
|---|---|---|---|
| WT (n=281) | 102 (99%) | 1 (0.97%) | 0 (0%) |
| *rmif-2* (n=364) | 141 (98.6%) | 2 (1.4%) | 0 (0%) |

| Chr V | SCO | DCO | TCO |
|---|---|---|---|
| WT (n=269) | 98 (98.98%) | 1 (1%) | 0 (0%) |
| *rmif-2* (n=362) | 134 (92.4%) | 9 (6.2%) | 2 (1.38%) |
| *rmh-1* (n=245) | 94 (95.9%) | 4 (4.08%) | 0 (0%) |

**Fig 7. RMIF-2 controls the CO position and suppresses the formation of additional COs.** (A) Schematic diagrams of chromosome (Chr.) IV (left) and V (right), showing the locations of the SNPs used in the PCR-based recombination assay. (B) Recombination frequencies on chromosomes IV (left) and V (right) assessed for different genetic intervals in WT, *rmif-2* and *rmh-1*. The 'theoretical' column is the expected recombination frequency based on the published genetic distance (http://www.wormbase.org). Statistical significance for recombination frequency over the total amount of worms was calculated using the Fisher's exact test: Chr. IV WT vs *rmif-2* ns (p = 0.568); Chr. V WT vs *rmif-2* ns (p = 0.4579); wt vs *rmh-1* ns (p = 0.6507); *rmif-2* vs *rmh-1* ns (p = 0.8662). Statistical significance of recombination frequencies between specific SNPs was calculated via a $\chi^2$ test: Chr IV Interval AB: WT vs Theoretical ns (p>0.05); WT vs *rmif-2* ** (p = 0.0048); Interval BC: WT vs Theoretical ns (p>0.05); WT vs *rmif-2* **** (p<0.0001); Interval CD: WT vs Theoretical ns (p>0.05); WT vs *rmif-2* ns (p = 0.2749). Chr V Interval AB: WT vs Theoretical ns (p>0.05); WT vs *rmif-2* ** (p = 0.0062); WT vs *rmh-1* **** (p<0.0001); *rmif-2* vs *rmh-1* * (p = 0.0362). Interval BC: WT vs Theoretical ns (p>0.05); WT vs *rmif-2* ns (p = 0.5759); WT vs *rmh-1* ns (p = 0.8938); *rmif-2* vs *mrh-1* ns (p = 0.6760). Interval CD WT vs Theoretical ns (p>0.05); WT vs *rmif-2* **** (p<0.0001); WT vs *rmh-1* **** (p<0.0001); *rmif-2* vs *rmh-1* ns (p = 0.1063). Interval DE: WT vs Theoretical ns (p>0.05); WT vs *rmif-2* ns (p = 0.0981); WT vs *rmh-1* ns (p = 0.0533); *rmif-2* vs *rmh-1* ns (p = 0.6122). Number of animals analyzed Chr IV: WT 281 worms, *rmif-2* 364 worms; Chr V: WT 269 worms, *rmif-2* 362 worms; *rmh-1* 245 worms. COs were shifted toward the chromosome center in the mutants compared with the WT. (C) The table contains the percentage of SNPs in each genetic interval on Chr IV (left) and Chr V (right). The number of COs per interval is shown in brackets. (D) Table displaying the number and percentage (in brackets) of single (SCO), double (DCO) and triple (TCO) crossovers in the genotypes analysed. *n* indicates the number of worms analyzed. $\chi^2$ test analysis showed that the change in crossover distribution between WT and *rmif-2* is significantly different on both chromosome IV (*** p = 0.0006) and chromosome V (** p = 0.0027). The change in crossover distribution between WT and *rmh-1* on chromosome V was statistically significant (**** p<0.0001). The change in crossover distribution between *rmif-2* and *rmh-1* on chromosome V was not statistically significant (p = 0.0995).

of the second X chromosome), and the presence of univalents in diakinesis nuclei. The rate of embryonic death was higher in the *rmh-1* mutant, possibly caused by more frequent random segregation of the univalents. In contrast, the reduced number of univalents in *rmif-2* mutants might be due to the higher prevalence of (usually rare) double and triple CO events on

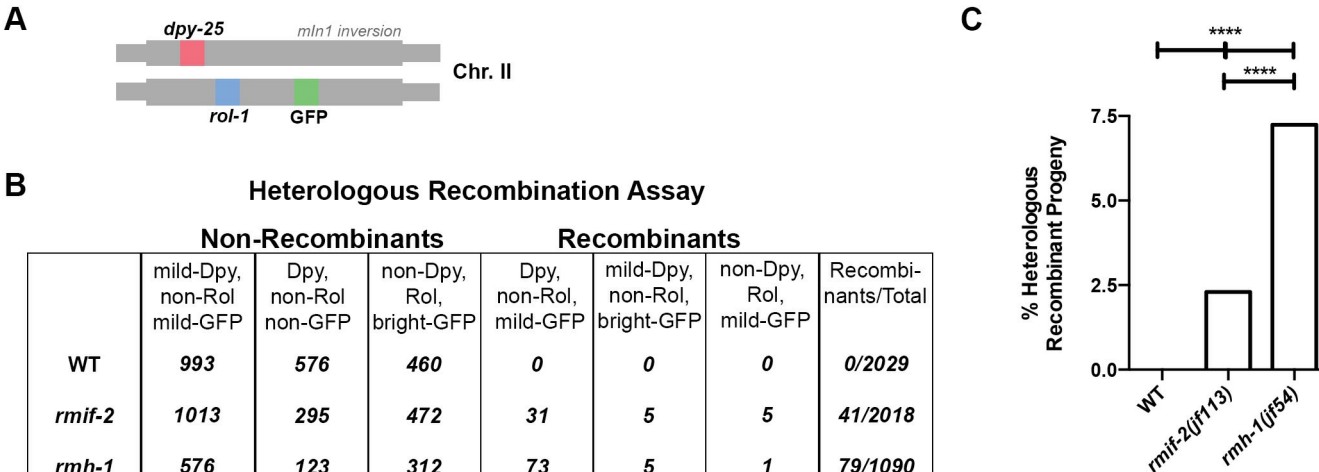

**Fig 8. RMH-1 and RMIF-2 suppress heterologous recombination to different extents.** (A) A heterologous recombination assay [12] was used to determine the involvement of *rmh-1* and *rmif-2* in suppressing illegitimate recombination events. The method used to score heterologous recombination relies on the use of the *mIn1* inversion on chromosome II (scoring for the exchange of shown genetic markers). (B) In WT (n = 2029 worms), no heterologous recombination was observed among the progeny; *rmif-2* (n = 2018) 41 recombinant progeny; *rmh-1* (n = 1090), 79 recombinant progeny. (C) Rate of heterologous recombinant progeny: WT, 0%; *rmif-2*, 2.3%; and *rmh-1*, 7.24%. The level of heterologous recombination in the *rmh-1* mutant is around three times higher than in the *rmif-2* mutant. Statistical analysis was done with a Fisher's exact test: WT vs *rmif-2* **** (p<0.0001); WT vs *rmh-1* **** (p<0.0001); *rmif-2* vs *rmh-1* **** (p<0.0001).

chromosomes. On the other hand, the average of 7 DAPI bodies in *rmif-2* could be the result of a premature dissociation of a bivalent after the CO designation, a phenotype also observed in the *rmh-1* mutant. The extra COs in *rmif-2* mutants might counteract the univalent formation, which likely arise through the absence of a pro-CO activity that is shared by RMH-1 and also seems to be lacking in *him-6* mutants [19,22,24]. Thus, the extra COs in *rmif-2* mutants might be linked to their lower rate of embryonic death compared with *rmh-1* mutants. The *C. elegans* genome encodes for two RMI1 homologs, namely RMH-1 and RMH-2 [19]. In fact, a degree of redundancy between RMH-1 and RMH-2 is indicated by the embryonic lethal phenotype of the double mutant, where no eggs hatch [19]. Nevertheless, RMH-1 seems to have evolved specialized non-shared meiotic activities, since *rmh-2* mutants did not display univalents in diakinesis nuclei and had a rate of larval lethality of only 7% [19].

In both *rmif-2* and *rmh-1* mutants, there was no difference in the total frequency of crossovers in comparison to the WT, however we detected a significant shift of COs away from chromosome arms toward the center.

Our data clearly indicate that the BTR complex must have activities that are not executed in the quantifiable recombination foci, where RMH-1 and RMIF-2 co-localization is mutually dependent. A striking difference between *rmh-1* and *rmif-2* mutants was their profiles of MSH-5 recombination foci. These foci were completely absent in the *rmh-1* mutant, where the MSH-5 signal was only detectable as a nuclear haze. In the *rmif-2* mutant, MSH-5 foci were significantly delayed and reduced in number; however, they co-localized with the ZHP-3 protein, which marks CO sites [50], and showed stringent co-localization with the COSA-1 pro-CO marker almost to WT levels. Therefore, a unique function of RMH-1 is to enrich MSH-5 in recombination foci. Moreover, the formation of joint chromosome structures in *rmh-1* partially depends on MSH-5 and to some extent on the unscheduled activity of NHEJ, so a future challenge will be to elucidate this RMH-1-specific activity. It seems as if MSH-5 is operating at recombination intermediates in the *rmh-1* mutant, but the levels are too low for detection or MSH-5 activity is not always found in foci.

The *rmif-2* and *rmh-1* mutants also differed in the extent of heterologous recombination in the *mIn1* inversion segment: the number of heterologous recombination events was three-fold lower in the *rmif-2* mutant compared with both *rmh-1* and *him-6* (this study and [12]). This strongly suggests the existence of meiotic BTR activity(ies) that do not strictly depend on the RMIF-2 protein. Alternatively, this might indicate that the BTR complex suppresses heterologous recombination via two independent activities, with only one of them depending on RMIF-2. These activities might include inhibiting the establishment of D-loops (which might be the source of the joint DNA structures) or decatenation by dissolution. It is not yet possible to determine which of these two possibilities is the more likely. Nevertheless, the strict interdependence of RMH-1 and RMIF-2 for localization into foci suggests that the *rmif-2*-independent activity of RMH-1 might relate to inhibiting D-loop formation, which might not involve the formation of distinct strong foci. The delayed formation and reduced numbers of MSH-5 and COSA-1 foci that we detected in *rmif-2* mutants argue that extra COs arising through the lack of such activities are not marked by MSH-5 and are likely class II COs, which are usually resolved by non-canonical resolvases.

Interestingly, the somatic cells of individuals with mutations in RMI2 display a "weak Bloom-like phenotype" [5]. For instance, elevated sister chromatid exchange is less pronounced in the chromosomes of RMI2 patients. This could be explained by residual dissolution activity mediated by Bloom in the absence of RMI2 and without a strict requirement for RMI2 to stabilize the BTR complex. The rate of heterologous recombination is lower in *rmif-2* mutants than in *him-6* or *rmh-1* mutants. Therefore, in meiosis RMIF-2 might not be involved in all HIM-6-mediated activities. It will be interesting to determine whether individuals carrying *rmif-2*/RMI2 mutations accumulate fewer genome rearrangements or translocations and whether this is associated with a lower risk of developing cancer compared with Bloom patients.

In summary, we have shown that the *C. elegans* RMI2 homolog RMIF-2 contributes to successful chromosome segregation in meiosis and shares numerous activities with RMH-1. However, specific RMIF-2-independent BTR activities were also identified in the germline, and the reduced susceptibility of *rmif-2* mutants to heterologous recombination might lead to fewer genome rearrangements and translocations than in mutants of the other BTR complex proteins. It will be interesting to investigate whether those observations made in the germline also hold true in the soma.

## Materials and methods

### Biochemical studies

Fractionated protein extracts for western blotting and immunoprecipitation assays were prepared as described previously [29]. A total of 50 μg protein from each fraction was used for western blotting, and at least 1 mg pooled soluble and insoluble nuclear fractions for immunoprecipitation assays. HA-tagged protein was immunoprecipitated using HA magnetic beads (Pierce #88836). For all immunoprecipitation experiments, beads were pre-equilibrated in buffer D (20% glycerol, 0.2 mM EDTA pH 8, 150 mM KCl, 20 mM Hepes-KOH (pH 7.9), and 0.2% Triton X-100, supplemented with protease inhibitor cocktail (Roche)) and incubated with the proteins overnight at 4°C with mild agitation. Beads were then washed three times in buffer D for 10 minutes, followed by acidic elution. For this, 25 μl glycine (100 mM, pH 2) was added to the beads and rotated for 5 mins. After magnetic separation of the beads, the supernatant (containing the target antigen) was neutralized with 5μl 1M Tris pH 9.2. Eluated proteins were separated on pre-cast 4–20% TGX gels (BioRad) in 1× SDS-Tris-glycine buffer and transferred onto nitrocellulose membrane for 1 hour at 4°C at 100V in 1× Tris-glycine buffer

containing 20% methanol. Membranes were blocked for 1 hour in 1× TBS containing 0.1% Tween (TBS-T) and 5% milk; primary antibodies were added in the same buffer and incubated overnight at 4˚C. Membranes were then washed three times in 1× TBS-T for 10 minutes and incubated with appropriate secondary antibodies in TBS-T containing 5% milk for 2 hours at room temperature. After another three washes, membranes were incubated with ECL (Euro-Clone) and developed using a ChemiDoc system (BioRad).

## Mass spectrometry analysis

Following co-immunoprecipitation, beads were washed with 50 mM ammonium bicarbonate and incubated with 200ng Lys-C protease for 4 h at 37˚C. The supernatant was collected, and bead-bound proteins and polypeptides were eluted twice with 100mM glycine pH 2. The pH of eluates was adjusted to alkaline by adding 1M TRIS pH 8 and disulfide bridges were reduced by adding dithiothreitol to a final concentration of 10mM and incubated at 45˚C for 30 min. Free thiols were alkylated with iodoacetamide to a concentration of 20mM and incubated at room temperature for 30min in the dark. Proteins were digested with 200ng trypsin overnight at 37˚C, and then acidified by adding TFA to a final concentration of 1%. Peptides were desalted on StageTips [51].

Peptide samples were separated on an Ultimate 3000 RSLC nano-flow chromatography system (Thermo Scientific, Dionex) using a pre-column for sample loading (Acclaim PepMap C18, 2 cm × 0.1 mm, 5 μm) and a C18 analytical column (Acclaim PepMap C18, 50 cm × 0.75 mm, 2 μm; both Thermo Scientific Dionex) with a linear 2–35% gradient of solvent B (80% acetonitrile, 0.1% formic acid; solvent A 0.1% formic acid) for 2 hours at a flow rate of 230 nl/min. Eluting peptides were analyzed on a Q Exactive HF Orbitrap mass spectrometer (Thermo Scientific). In data-dependent mode, survey scans were acquired in a mass range of 380–1650 *m/z* with lock mass on at a resolution of 120,000 at 200 m/z. The AGC target value was set to 3E6 with a maximal injection time of 60 ms. The 10 most intense ions were selected with an isolation width of 2 *m/z*, and fragmented in the HCD cell with a normalized collision energy of 28%. Spectra were recorded at a target value of 1E5 with a maximal injection time of 250 ms and a resolution of 30,000. Peptides with an unassigned charge state or a charge of +1 or greater than +6 were excluded from fragmentation. The peptide match feature was set to preferred and the exclude isotope feature was enabled. Selected precursors were dynamically excluded from repeated sampling for 30 s. Raw data were processed using the MaxQuant software package 1.6.17.0 (http://www.maxquant.org/) [52] and searched against a *C. elegans* reference database (Wormbase, version WS269) and a custom database of common contaminants. The search was performed with full tryptic specificity and a maximum of two missed cleavages. Carbamidomethylation of cysteine residues was set to fixed and oxidation of methionine, and N-terminal protein acetylation as variable modifications—all other parameters were set to default. The "match between runs" feature was enabled, label-free quantification (LFQ) activated but without normalization. Results were filtered at protein and peptide level for a false discovery rate of 1%. The protein-group table was further processed in R as follows (R Core Team (2020). R: A language and environment for statistical computing. R Foundation for Statistical Computing, Vienna, Austria. https://www.R-project.org/.: data were filtered for reverse hits, contaminants and proteins "only identified by site". Raw protein group intensity values were median normalized per sample. After filtering for at least two valid quantification values in at least one group missing values were replaced by drawing random values from a normal distribution at the detection limit. The LIMMA package was used for statistical comparison, using a paired design to account for batch effects between the three independent replicates [53]. Peptide spectrum matches (PSM) in Table 1 correspond to "MS/MS counts" as

defined in the MaxQuant output, representing PSM that were considered by the MaxLFQ algorithm. For the complete list of peptides see, **S1 File.**

## Bioinformatic analysis of Y104H12D.4

The RMI2 protein family is highly conserved in the plant and animal kingdoms. In some nematodes, such as *Trichinella pseudospiralis*, RMI2 family members can be identified as significant hits using hmmsearch with the hidden Markov model (HMM) of the Pfam RMI2 domain (UniProt|A0A0V0YD80, E-value 6.2e-05), [54,55]. However, no Pfam RMI2 hit was identified in the *C. elegans* proteome. Likewise, one significant hit to the RMI2 domain was found in the termite *Zootermopsis nevadensis* (UniProt|A0A067R4M9, E-value 4.2e-21) but none in *Drosophila melanogaster*. The Pfam RMI2 domain belongs to the OB-fold clan, which comprises 107 domains with a wide range of molecular functions, including oligonucleotide or oligosaccharide binding and protein–protein interactions. The common structural feature of the OB-fold is a five-stranded beta-sheet forming a closed beta-barrel [56].

As none of the RMH-1 co-purifying proteins had a significant hit to OB-fold superfamily domains in a Pfam hmmsearch, we applied the HMM-HMM comparison with HHPRED to search for remote homologs [57]. RMIF-2 (Y104H12D.4) was the only candidate with a predicted OB-fold, where the best hits in the PDB structural database were to the yeast RFA2 (Replication factor A protein 2; 6I52_B, probability 74.5%) and human RMI2 (3MXN_B, probability 72.5%). In the Pfam database, the best hit was to the RMI2 domain (probability 66.24%). The hits covered nearly the complete length of RMIF-2 (residues 5–101), including the five-stranded beta-sheet. RMIF-2 orthologs can be identified in other nematodes using NCBI blastp searches of the NCBI non-redundant protein database [58], but not in nematodes with RMI2 orthologs. Since nematode RMI2 orthologs are highly related to RMI2 protein family members, it is uncertain whether RMIF-2 is an RMI2 ortholog. However, the mutually exclusive distribution of RMI2 and RMIF-2 orthologs in nematode taxa and the structural similarity to the OB-fold superfamily, in addition to biochemical and genetic data, are strong indications that RMIF-2 is indeed a functional RMI2 homolog.

In the case of RMIF-2 (UniProt accession Q8MXU4), the conservation histogram and the consensus sequence are based on an alignment of nematode orthologs, and for RMI2 (Q96E14) we used a wide selection of eukaryotic orthologs, including animal and plant sequences. Secondary structure elements were predicted by Jpred, where the helices are marked as red tubes, and sheets as green arrows, [59]. The visualization was performed as, [60].

For the ribbon diagram of the human RMI core complex and a model of the putative *C.elegans* RMIF-2 OB-fold the 3D coordinates of the crystal structure of RMI1 and RMI2 were retrieved from the RCSB PDB protein database, [61,62]. The model of the RMIF-2 OB-fold was created with MODELLER, [63] based on an alignment performed by the remote homology detection and 3D structure prediction server HHpred, [64]. In this search, a multiple alignment of RMIF-2 orthologs (including region 18–89 of *C.elegans* RMIF-2) was compared with profile hidden Markov models (HMMs) of sequences from the PDB structure database and the best, but not significant, hit was to RMI2 (region 58–138). The model was aligned to the RMI2 coordinates and visualized with pymol (https://pymol.org/2/).

## Worm strains

All worms strains were grown at 20˚C using standard techniques [65] on Nematode Growth Medium seeded with *Escherichia coli* OP50. The N2 Bristol strain was used as the WT

reference. Unless otherwise stated, prepicked L4 hermaphrodite worms grown at 20˚C for 16–24 hours were used for all experiments.

The following published mutant alleles and tagged lines were used in this study: *rmh-1(jf54) I* [19] (UV173), *him-6(ok412) IV* [24] (VC193), *spo-11(ok79)/nT1[unc-?(n754) let-?(m435)] (IV; V)* [66] (AV106), *jfSi38 [gfp::rmh-1 cb-unc-119(+)] II* [19] (UV208), *msh-5(me23)/nT1 [unc-?(n754) let-?] (IV;V)* (*Caenorhabditis Genetics* Center) (AV115), *rmh-1(jf172 [ha::rmh-1]) I* [30] (NSV240), *[ollas::cosa-1] III* [30] (NSV97), *[gfp::msh-5] IV* [30] (NSV129), *cosa-1 (tm3298)/qc1[qLs26] III* [39] (AV590), *Hawaiian CB4856* (*Caenorhabditis* Genetics Center), *rmh-1(jf54) I in CB4856* [19] (UV223), and *mIn1[mIs14 rol-1(e91)]/dpy-25(e817) II* [12] (DW579), *cku-70(tm1524) III* (FX1524) (*Caenorhabditis Genetics* Center).

The following mutant alleles and tagged lines were created for this study: *rmif-2(jf113)/ tmC25 [unc-5 (tmIs1241)] IV* (UV193), *rmif-2(jf139 [rmif-2::ha]) IV* (UV194), *rmif-2(jf186 [rmif-2::3xflag]) IV* (UV209), *rmh-1(jf172 [ha::rmh-1]) I; rmif-2(jf186 [rmif-2::3xflag]) IV* (UV210), *jfSi38 [gfp::rmh-1 cb-unc-119(+)] II; rmif-2(jf139 [rmif-2::ha]) IV* (UV195), *rmif-2 (jf113) IV spo-11(ok79)/nT1 [unc-?(n754) let-?(m435)] (IV;V)* (UV196), *rmh-1(jf54)/hT2 I; rmif-2(jf113) IV* (UV211), *top-3(jf110 [top-3::ollas]) III* (UV212), *top-3(jf110 [top-3::ollas] III; rmif-2(jf113) IV* (UV199), *rmif-2(jf113)) /tmC25 [unc-5(tmIs1241)] him-6(jf93 [him-6::ha]) IV* (UV213), *jfsi38 [gfp::rmh-1; cb-unc-119(+)] I; rmif-2(jf113) IV* (UV214), *[ollas::cosa-1] III; rmif-2(jf113)/tmC25 [unc-5(tmIs1241)] IV* (UV215), *rmh-1(jf54)I; [ollas::cosa-1] III* (UV230), *rmif-2(jf113) [gfp::msh-5] IV* (UV216), *rmh-1(jf54) I; [gfp::msh-5] IV* (UV217), *him-6(ok412) [gfp::msh-5] IV* (UV231), *top-3(jf101) [Y56A3A.27::unc-119(+)]/hT2 (I;III); [gfp::msh-5] IV* (UV232), *cosa-1(tm3298)/qc1[qLs26] III; rmif-2(jf113) IV* (UV224), *rmh-1(jf103 [ha::degron:: M01E11:3]) I; unc-119(ed3) III* (UV228), *rmh-1(jf54) I;msh-5(me23)/nT1 (IV;V)* (UV227), *rmif-2(jf113) him-6(ok412)* (UV218), *rmh-1(jf172 [ha::rmh-1]) I; rmif-2(jf113)/tmC25 [unc-5 (tmIs1241)] IV* (UV219), *rmh-1(jf54) I; rmif-2(jf139[rmif-2::ha]) IV* (UV225), *cosa-1(tm3298)/ qC1[qIs26] III; rmif-2(jf139 [rmif-2::ha]) IV* (UV230), *rmif-2(jf113) in CB4856* (UV222), *mIn1 [mIs14 rol-1(e91)]/dpy-25(e817) II; rmif-2(jf113) IV* (UV221), and *mIn1[mIs14 rol-1(e91)]/ dpy-25(e817) II; rmh-1(jf54) I* (UV220); *rmh-1(jf54) I; cku-70(tm1524) III; msh-5(me23) IV/ nT1(IV,V)* (UV236); *rmh-1(jf54) I; cku-70(tm1524)III; rmif-2(jf113) IV/hT2(I,III)* (UV235).

**CRISPR-Cas9.** All strains generated by CRISPR-Cas9 were confirmed by sequencing and backcrossed to WT worms twice prior to use. Strains were generated using a published protocol [67]. Tagged lines had WT levels of viability, hatch rates and lack of males in their progeny (Table 2).

**Generation of *rmif-2::ha*.** We tagged the endogenous *rmif-2* locus at the protein C-terminus with an HA-tag and a 5x-Gly linker sequence. The repair template (from Integrated DNA Technologies, 4 nmole Ultramer DNA Oligo) was composed of 35-bp homology to the *rmif-2* sequence, into which the HA sequence was inserted.

The following guide RNA was used: crRNA (from Dharmacon, Edit-R CRISPR-Cas9 Synthetic crRNA 20 nmol), 5′ AGAGATGATCAGTTGGCTGT 3′. The sequence of the repair template was: 5′ GCG AAA AAA AAA TTA GAG ACG CAG ACG ATG ACG GAG AGA TGA TCA AGC GTA ATC TGG AAC ATC GTA TGG GTA TCC TCC TCC TCC TCC GTT GGC TGT TGG TGA GAT GAT CAC TGA AAA TTG GAA ATA AAT TTG AAG 3′.

**Generation of *top-3::ollas*.** The OLLAS sequence (5' AGC GGT TTT GCT AAC GAA CTG GGT CCC CGC TTG ATG GGA AAG 3') was inserted into an internal location (corresponding to between Gly 635 and Gly 636). The repair template (from Integrated DNA Technologies, 4 nmole Ultramer DNA Oligo) was composed of 35-bp homology to the *top-3* sequence. The following guide RNA was used: crRNA (from Dharmacon, Edit-R CRISPR-Cas9 Synthetic crRNA 20 nmol), 5′ CCT GGA GGT GGT GGT GGG GGA GG 3′. The sequence of the repair template was 5′ GGT GGA GGC CCA CCA AGA GGA CCT GGA

GGT GGT GGT AGC GGT TTT GCT AAC GAA CTG GGT CCC CGC TTG ATG GGA AAG GGG GGA GGC CCT ACA GGC CCG CCG GCT CCT CCA AA 3′.

**Generation of *rmif-2::3xflag*.**   We tagged the endogenous *rmif-2* locus at the protein C-terminus with a 3×FLAG-tag and a 5x-Gly linker sequence. The 3×FLAG sequence was inserted into a repair template (from Integrated DNA Technologies, 4 nmole Ultramer DNA Oligo) with 35-bp homology to the *rmif-2* sequence. The guide RNA was: crRNA (from Dharmacon, Edit-R CRISPR-Cas9 Synthetic crRNA 20 nmol), 5′ AGAGATGATCAGTTGGCTGT 3′. The sequence of the repair template was: 5′ AAA AAA TTA GAG ACG CAG ACG ATG ACG GAG AGA TGA TCA CTT GTC ATC GTC ATC CTT GTA ATC GAT ATC ATG ATC TTT ATA ATC ACC GTC ATG GTC TTT GTA GTC TCC TCC TCC TCC TCC GTT GGC TGT TGG TGA GAT GAT CAC TGA AAA TTG AAA TAA AAT TTG AAG 3′.

**Generation of *ha::degron::rmh-1*.**   We tagged the endogenous *rmh-1* locus with a 5′ HA-degron tag using CRISPR/Cas9. The HA and degron sequences were inserted into a repair template (from Integrated DNA Technologies, 4 nmole Ultramer DNA Oligo) with 35-bp homology to the *rmh-1* sequence. We were not successful in obtaining efficient protein degradation using the degron sequence. The guide RNA was: crRNA (from Dharmacon, Edit-R CRISPR-Cas9 Synthetic crRNA 20 nmol), 5′ AACTTGATCGTCTCTTTTCA 3′. The sequence of the repair template was: 5′ TTG CAG AGC GAA CGC ATA TAA AAA CTA CAA AAT ATA TGT ACC CAT ACG ACG TCC CAG ACT ACG CCA TGC CTA AAG ATC CAG CCA AAC CTC CGG CCA AGG CAC AAG TTG TGG GAT GGC CAC CGG TGA GTC ATA CCG GAA GAA CGT GAT GGT TTC CTG CCA AAA ATC AAG CGG TGG CCC GGA GGC GGC GGC GTT AGT GAA GAT GAA AGA AAC TGA ACT TGA TCG TCT CTT TTC ATG GCT TGC TAG GAA ACA TTA CCC ATT CAA GAG AGA ATG 3′.

**Deletion of the *rmif-2* locus.**   To generate a full deletion of the *rmif-2* locus, two crRNAs were designed to target the beginning and end of the *rmif-2* gene. A repair template (from Integrated DNA Technologies, 4 nmole Ultramer DNA Oligo) containing the 5′-UTR and a STOP codon was designed and synthesized. The 3357 base pairs deletion was confirmed by sequencing. The guide RNAs used were: crRNA1, 5′ TGATAGTTTCTCCGGTGCAG 3′; and crRNA2, 5′ ATGACGGAGAGATGATCAGT 3′ (from Dharmacon Edit-R CRISPR-Cas9 Synthetic crRNA 20 nmol). The sequence of the repair template was: 5′ CGC GAT ACT TGC ACA ATC GTC TCG ATC GCA CAT TTT CTA TGG ATT TTC CGG TTT TTT GGG GTA AAA AAT GGG TGA AAA TAG GTA AAA AAA AGC CGG AAT AAA CCG AGA GAT TTT GAA GTT TTC GAG GAA GCA GAG AAA CAG AGA AAT TTA GAA AAA AAC AAA AAA ACA TTT TTG CGA AAA AAA AAT TAG AGA CGC AGA CG 3′.

## Viability analysis

Single L4-stage worms were transferred to plates. The worms were picked individually and moved onto new plates every 24 hours for 4 days. Dead eggs and viable larvae were scored 24 hours after the mother was removed, and male progeny were counted 3 days later. The viability of embryos was calculated as the number of hatched eggs divided by the total number of eggs laid, and percentage of males was calculated as the total number of male progeny divided by the number of hatched eggs.

## Immunofluorescence analysis

Immunocytological analysis was performed as previously described [68]. L4 hermaphrodite worms were incubated at 20˚C for 20–24 hours. Their gonads were then dissected in 1× PBS on Superfrost slides, fixed in final 1% paraformaldehyde for 5 min at room temperature, and frozen in liquid nitrogen. After freeze-cracking and fixation in ice-cold methanol at −20˚C for

10 minutes, the slides were washed three times in PBS-T (1× PBS, 0.1% Tween) at room temperature for 10 minutes. Non-specific binding sites were blocked by incubation in PBS-T containing 1% BSA for 0.5–1 hour. Primary antibody diluted in PBS-T was applied to slides and incubated overnight at 4°C in a dark, humid chamber. Slides were then washed three times in PBS-T at room temperature for 10 minutes and incubated with secondary antibody diluted in PBS-T for 2 hours at room temperature in a dark, humid chamber. Slides were again washed three times for 10 minutes in PBS-T. Slides were then incubated with DAPI (60μl of a 2 μg/ml stock solution diluted 1:1000 in water) for 1 minute at room temperature, washed with PBS-T for 30 minutes at room temperature, and mounted with Vectashield Mounting Medium (Vector Labs #H-1000).

For detection of GFP::MSH-5, gonads were dissected and fixed in 1× EGG buffer containing 0.1% Tween instead of PBS-T.

The following antibodies dilutions were used in immunolocalization studies: mouse anti-HA (1:100; Cell Signaling), mouse monoclonal anti-GFP (1:500; Roche), rabbit anti-RAD-51 (1:500; a gift from the Zetka laboratory), rabbit polyclonal anti-OLLAS (1:1500; GenScript), rabbit polyclonal anti-HIM-8 (1:10,000; Novus), guinea pig polyclonal anti-HTP-3 (1:500; a gift from the Zetka laboratory), chicken polyclonal anti-SYP-1 (1:500; a gift from the Martinez-Perez laboratory), and guinea pig polyclonal anti-ZHP-3 (1:250; a gift from the Bhalla laboratory). All secondary antibodies were Alexa Fluor conjugated and used at a 1:400 dilution.

The following antibodies were used at the indicated dilutions for western blot analysis: mouse polyclonal anti-FLAG (1:1000; Sigma), mouse monoclonal anti-HA (1:1000; Cell Signaling), mouse monoclonal anti-GAPDH (1:5000; Ambion), guinea pig polyclonal anti-lamin (1:10,000; a gift from the Krohne laboratory), and mouse monoclonal anti-tubulin (1:2000; Thermo Fisher), rat anti-HA-Peroxidase (1:2000, Roche), mouse HRP anti-DDDDK tag (1:10000, Abcam). HRP-conjugated secondary antibodies were used as follows: anti-mouse (1:2500; Cell Signaling), and anti-guinea pig (1:5000; Abcam).

**Quantification of nuclear foci.** For quantification of COSA-1, MSH-5, and RAD-51 foci, the hermaphrodite gonad was divided into seven equal zones (in Adobe Photoshop) from the mitotic tip to late pachynema. The number of foci per nucleus was counted in each zone, in at least three gonads per genotype. RAD-51 quantification graphs show the percentage of nuclei corresponding to each of the following categories: 0 foci, 1 focus, 2–3 foci, 4–6 foci, 7–12 foci, and >12 foci per nucleus. For the complete statistics of RAD-51 foci quantification, see **S2 File**.

For COSA-1/ZHP-3 co-localization in foci, a similar system was used but OLLAS::COSA-1 foci were scored in each nucleus in zone 7 only (late pachynema) for co-localization with ZHP-3 signals.

For quantification of TOP-3 foci, the hermaphrodite gonad was divided into four equal zones from the onset of meiosis in the transition zone to late pachynema (Adobe Photoshop) and the number of foci per nucleus was counted in each zone, in three gonads per genotype.

## Length of the mitotic zone

The length of the mitotic zone in WT and mutant strains was measured by counting the number of cell rows in the hermaphrodite gonad from the mitotic tip to the transition zone, where chromatin adopts a half-moon shape. Cell rows were counted in seven WT worms and ten *rmif-2* worms.

## X chromosome pairing

To assess X chromosome pairing, the number of nuclei containing one (i.e. paired) or two (i.e. unpaired) HIM-8 foci was recorded in each of seven zones (starting from the mitotic tip to late

pachynema) of the hermaphrodite gonad. Graphs show the percentage of paired HIM-8 nuclear signals in each zone.

## Quantification of Synaptonemal complex assembly

To assess synaptonemal complex assembly, the number of nuclei with full co-localization between HTP-3 (chromosome axis marker) and SYP-1 (marker of the central element of the synaptonemal complex) were quantified. The hermaphrodite gonad was divided in seven equal zones from the mitotic tip to late pachynema. Graphs show the percentage of nuclei with complete synapsis in each zone.

## Recombination assays

Recombination frequencies and CO localization were assessed in the WT, *rmif-2(jf113)*, and *rmh-1(jf54)* strains by determining the differences in unique SNPs for chromosomes IV and V between the N2 Bristol and Hawaiian (Hw) strains. For this, mutants were crossed to the Hw strain to generate a mutant with introgressed Hw chromosomes IV and V. Subsequently, *rmif-2* mutant males in Hw and *rmif-2* mutant hermaphrodites in WT were crossed to generate F1 *rmif-2* mutants heterozygous for Hw on chromosomes IV and V and recombination events took place in F1 worms. F1 worms were mated with WT worms expressing a tomato transgene to introduce a WT paternal chromosome and enable recombination events in oogenesis to be monitored. After laying eggs, F1 parent worms were lysed and genotyped to ensure that the first mating worked, using the presence of Hawaiian SNPs as a read-out for chromosomes IV and V. Single F2 hermaphrodites were lysed and analyzed by PCR (restriction by DraI). SNP positions and primers are shown in Table 3.

## Heterologous recombination assay

To examine illegitimate recombination events between heterologous sequences, we used the heterologous recombination assay, as previously described, with the *dpy-25/mIn1* [*rol-1* GFP]

**Table 3. SNP positions and primers.**

| SNP identifier | SNP position | Primer pair |
|---|---|---|
| Chromosome IV | | |
| A | -16 | Forward: 5′ CGCATAAATCCAACGTTCTCTG 3′<br>Reverse: 5′ AATCCATAAGTTTCGTGTTGG 3′ |
| B | 1 | Forward: 5′ AAAATGGGAAGCGTACCAAA 3′<br>Reverse: 5′ TGCTTGTAGCGTTTCCAAGA 3′ |
| C | 8 | Forward: 5′ GACACGACTTTAGAAACAACA 3′<br>Reverse: 5′ TGGTATGGAGTCCCTATTTTG 3′ |
| D | 14 | Forward: 5′ GAATTTCAGGTGTTGGAAGG 3′<br>Reverse: 5′ TGCTCTGAAAAAATTGGCTG 3′ |
| Chromosome V | | |
| A | -17.5 | Forward: 5′ TTTCGGAAAATTGCGACTGT 3′<br>Reverse: 5′ CGCGTTTTGGAGAATTGTTT 3′ |
| B | -5 | Forward: 5′ GAGATTCTAGAGAAATGGACACCC 3′<br>Reverse: 5′ AAAAATCGACTACACCACTTTTAGC 3′ |
| C | 1 | Forward: 5′ AGAAATGATCCGATGAAAAGC 3′<br>Reverse: 5′ CCGATAGTGTTCATAGCATCCC 3′ |
| D | 5.8 | Forward: 5′ AGCCACATAAGCGCAATAAC 3′<br>Reverse: 5′ GTGACGGAACAAACTCATCTGC 3′ |
| E | 17.8 | Forward: 5′ GCCACTGATGGGACAGAACC 3′<br>Reverse: 5′ CAGAAAAATTGCCAAAACTACCG 3′ |

II *C. elegans* strain [12], in which one copy of chromosome II is marked with the semi-dominant *dpy-25* mutation and the second copy contains the *mIn1* inversion (marked with the recessive *rol-1* mutation) and a semi-dominant GFP-expressing transgene. In the absence of recombination across the inversion, progeny of the heterozygous *dpy-25/mIn1* [*rol-1* GFP] II hermaphrodite would produce 50% heterozygous *dpy-25/mIn1* (mild-Dpy, non-Rol, mild-GFP), 25% homozygous *dpy-25/dpy-25* (Dpy, non-Rol, non-GFP), and 25% homozygous *mIn1/mIn1* (non-Dpy, Rol, bright-GFP). CO between the *mIn1* and normal versions of chromosome II would lead to different combinations of the phenotypes. Possible phenotypes of recombinant progeny include: Dpy, mild-GFP; mild-Dpy, bright-GFP; Rol, mild-GFP; Dpy, Roll, bright-GFP; Dpy, bright-GFP; Dpy, Rol, mild-GFP; Dpy, Rol; mild-Dpy, Rol, bright-GFP; mild-Dpy, Rol, mild-GFP; mild-Dpy, Rol; Rol; mild-GFP; bright-GFP; and non-Dpy, non-Rol, non-GFP. We crossed our *rmif-2* and *rmh-1* mutant strains into the *mIn1* strain and recorded illegitimate recombination events in the progeny. We scored 2029 WT worms, 993 were mild-Dpy, non-Rol, mild-GFP; 576 were Dpy, non-Rol, non-GFP; and 460 were non-Dpy, Rol, bright-GFP. No heterologous recombination was detected in the progeny (0%). Of the progeny of 2018 *rmif-2* mutant worms, 1013 were mild-Dpy, non-Rol, mild-GFP; 295 were Dpy, non-Rol, non-GFP; and 472 were non-Dpy, Rol, bright-GFP. The numbers of recombinant progeny were 31 Dpy, non-Rol, mild-GFP; five mild-Dpy, non-Rol, bright-GFP; and five non-Dpy, Rol, mild-GFP, corresponding to 2.03% of total recombinants. Of the progeny of 1090 *rmh-1* mutant worms, 576 were mild-Dpy, non-Rol, mild-GFP; 123 Dpy, non-Rol, non-GFP; and were 312 non-Dpy, Rol, bright-GFP. The numbers of recombinant progeny were 73 Dpy, non-Rol, mild-GFP; five mild-Dpy, non-Rol, bright-GFP; and one non-Dpy, Rol, mild-GFP, corresponding to 7.24% of total recombinants.

### Image acquisition

All microscopy experiments were done using a DeltaVision Epifluorescence Microscope system with 1.3 Megapixel CCD camera, 7-color LED for fluorescence, white LED for transmitted light (UPlanSApo 100×/1.40 oil immersion objective lens) with softWoRx suite R6.1.1 image analysis deconvolution software (Applied Precision); ImageJ (National Institutes of Health), and Adobe Photoshop software. Unless otherwise stated, images are maximum projections of entire nuclei. Images acquired with the DeltaVision were deconvolved using the softWoRx deconvolution algorithm. Maximum intensity projections of deconvolved images were generated using Fiji/ImageJ after background subtraction using a rolling ball radius of 50 pixels. Images of gonads consist of multiple stitched pictures that were processed in the same manner. This is necessary, due to the size limitation of the field of view at high magnification. Stitching of images to build up entire gonad was performed manually in Adobe Photoshop.

### Statistical analysis

Statistical analyses were performed using GraphPad Prism 6 (GraphPad) software and Microsoft Excel. Fisher's exact tests, Student T-tests, $\chi^2$ tests, Mann-Whitney tests mean and standard deviation, and statistically significant differences are shown in figures and reported in figure legends. *p* values of below 0.05 were considered statistically significant: * $p < 0.05$, ** $p < 0.01$, *** $p < 0.005$, and **** $p < 0.0001$.

### Supporting information

**S1 Fig. A Ribbon diagram of the human RMI core complex and a model of the putative *C. elegans* RMIF-2 OB-fold.** (A) 3D coordinates of the crystal structure of RMI1 (in green) and RMI2 (in grey) were retrieved from the RCSB PDB protein database [61,62]. A model of the

RMIF-2 OB-fold is shown in cyan and was created with MODELLER, [63], based on an alignment performed by the remote homology detection and 3D structure prediction server HHpred, [64]. In this search, a multiple alignment of RMIF-2 orthologs (including region 18–89 of *C.elegans* RMIF-2) was compared with profile hidden Markov models (HMMs) of sequences from the PDB structure database and the best, but not significant, hit was to RMI2 (region 58–138). The model was aligned to the RMI2 coordinates and visualized with pymol (http://www.pymol.org/). Two views on the ensemble are given, rotated by 180 degrees.
(TIF)

**S2 Fig. X chromosome pairing, meiotic entry and synapsis in the *rmif-2* mutant.** (A) HIM-8 staining (in green) was used to follow X chromosome pairing in *rmif-2*. Scale bar, 10 μm. Bottom: percentage of nuclei with a paired HIM-8 signal. Gonads were divided into seven equal zones from the mitotic tip to late pachynema (n = 3 gonads per genotype). X chromosome pairing was significantly slower in *rmif-2* than in the WT but reached WT levels in mid/late pachynema. (B) Length of the mitotic zone in WT and *rmif-2*. The mitotic zone is slightly extended in the mutant: 23.4 (±3.0 SD) cell rows in *rmif-2* vs 20.4 (±2.6 SD) cell rows in the WT (n = 7 WT gonads; n = 10 *rmif-2* gonads). Significant differences in foci distribution were determined using a Student T-test: ns, not significant ($p > 0.05$), * $p < 0.05$, **** $p < 0.0001$. (C) Staining for HTP-3 in cyan (chromosome axis) and SYP-1 protein in yellow (central element of the synaptonemal complex) served as a read-out for the kinetics of SC assembly in the WT and *rmif-2* mutant. Scale bar, 10 μm. Bottom: percentage of nuclei with complete synapsis. Gonads were divided into seven equal zones from the mitotic tip to late pachynema (n = 3 gonads per genotype). No major defects in synapsis were apparent in the *rmif-2* mutant. Significant differences were determined using a Student T-test: ns, not significant ($p > 0.05$).
(TIF)

**S3 Fig. Diakinesis chromosome counts in several mutants and nuclear RMIF-2 localization in *cosa-1*.** (A) Quantification of diakinesis DAPI-stained bodies in -1 oocytes in the WT and in *rmif-2(jf113)*, *rmh-1(jf54)*, *cosa-1(tm3298)*, *cosa-1(3298); rmif-2(jf113)*, *msh-5(me23)*, *rmh-1(jf54); msh-5(me23)*, *rmh-1(jf54); cku-70(tm1524); msh-5(me23)* and *cku-70(tm1524)* mutants. Numbers of DAPI-stained bodies diakinesis were scored in the WT (n = 32 oocytes), *rmif-2* (n = 41), *rmh-1* (n = 74), *cosa-1* (n = 17), *cosa-1; rmif-2* (n = 29), *msh-5* (n = 13), *rmh-1; msh-5* (n = 28), *rmh-1(jf54); cku-70(tm1524); msh-5(me23)* (n = 26) and *cku-70(tm1524)* (n = 19). Data are the mean and standard deviation (error bars). Significant differences were determined using a Student T-test: ns, not significant ($p > 0.05$); **** $p < 0.0001$. (B) Representative diakinesis nucleus of the *cosa-1(tm3298)*, *cosa-1(tm3298); rmif-2(jf113)*, *cku-70(tm1524)*, *msh-5(me23)*, *rmh-1(jf54); msh-5(me23)* and *rmh-1(jf54); cku-70(tm1524); msh-5(me23)* genotypes. Scale bars: 10 μm. (C) Representative image of mid and late pachynema nuclei stained with DAPI (in magenta) and HA (in green). RMIF-2 localizes to bright foci throughout mid and late pachynema. In the *cosa-1* mutant, RMIF-2 fails to localize into nuclear foci, and only a few small foci are detected in the cytoplasm. Scale bars: 10 μm.
(TIF)

**S1 File. Complete list of RMH-1 interacting proteins as determined by affinity purification mass spectrometry.**
(XLSX)

**S2 File. Raw data and statistical analysis for the RAD-51 foci quantification as determined by a Fisher's exact test.**
(XLSX)

**S3 File. Raw data.**
(XLSX)

## Acknowledgments

The authors thank Angela Graf for outstanding technical assistance, and Mona Frey and Marlene Jagut for generating *ha*::*degron*::*rmh-1*. We thank all members of the Jantsch lab for helpful discussions and support as well as Josef Roehsner for the help with *rmif-2*::*3xflag*. Proteomics analyses were performed at the Mass Spectrometry Facility, Max Perutz Labs, using the Vienna BioCenter Core Facilities instrument pool, with special help of Dorothea Anrather. We thank Monique Zetka, Needhi Bhalla, and Enrique Martinez-Perez for sharing reagents and the Max Perutz Labs BioOptics facility (Irmgard Fischer, Josef Gotzmann and Thomas Peterbauer) for trainings and image acquisition support.

## Author Contributions

**Conceptualization:** Maria Velkova, Nicola Silva, Maria Rosaria Dello Stritto, Verena Jantsch.

**Data curation:** Maria Velkova, Maria Rosaria Dello Stritto.

**Formal analysis:** Maria Velkova, Maria Rosaria Dello Stritto, Alexander Schleiffer, Pierre Barraud, Markus Hartl.

**Funding acquisition:** Verena Jantsch.

**Investigation:** Maria Velkova, Nicola Silva, Maria Rosaria Dello Stritto, Markus Hartl.

**Project administration:** Verena Jantsch.

**Supervision:** Verena Jantsch.

**Writing – original draft:** Maria Velkova, Verena Jantsch.

**Writing – review & editing:** Nicola Silva, Maria Rosaria Dello Stritto.

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
