## [Decision Letter · Decision Letter 0]

8 Mar 2021

Dear Dr. Jantsch,

Thank you very much for submitting your Research Article entitled 'Characterization of the Caenorhabditis elegans RMI2 functional homolog-2 (RMIF-2) reveals similarities and differences to RMH-1 (RMI1) within the BTR complex in meiosis' to PLOS Genetics.

The manuscript was fully evaluated at the editorial level and by independent peer reviewers. You will see that all three reviewers recognized the importance of this work. However, they raised substantial concerns about the lack of statistical analysis and asked for additional insights into the function of RMIF-2, distinct from other members of the BTR complex. Based on the reviews, we will not be able to accept this version of the manuscript, but we would be willing to review a much-revised version. We cannot, of course, promise publication at that time.

Should you decide to revise the manuscript for further consideration here, your revisions should address the specific points made by each reviewer. In particular, Reviewers #1 and #2 ask to provide further insight into the rmif-2; him-6 phenotype and have suggestions on the quantification of cytological data. I realize that the genetic interaction between rmh-1 and him-6 was shown in the previous work (Jagut et al., 2016). However, as Reviewer #1 also noted, it would be beneficial to include the analysis of rmi-1 for a direct comparison. Reviewer #3 has questions about the repair outcome in rmif-2 single and rmh-1; rmif-2 double mutants, which seem addressable by minor experimentation and text revisions.

If you decide to revise the manuscript for further consideration at PLOS Genetics, please aim to resubmit within the next 60 days, unless it will take extra time to address the concerns of the reviewers, in which case we would appreciate an expected resubmission date by email to plosgenetics@plos.org.

We are sorry that we cannot be more positive about your manuscript at this stage. Please do not hesitate to contact us if you have any concerns or questions.

Yours sincerely,

Yumi Kim

Guest Editor

PLOS Genetics

Gregory P. Copenhaver

Editor-in-Chief

PLOS Genetics

Reviewer's Responses to Questions

**Comments to the Authors:**

Reviewer #1: The manuscript by Velkova et al., describes the identification and functional characterization of the C. elegans ortholog of RMI2 (RMIF-2), a member of the BTR complex critical for recombination processing. The authors isolated RMIF-2 as an interactor of RMI1 (RMH-1), another member of BTR, through pull downs; RMIF-2 contains an OB fold, a key feature of the mammalian protein. Analysis of steady state protein levels and localization reveal that similar to the mammalian homolog, RMIF-2 is required for the stability of RMH-1 and for the recruitment of the other members of the BTR complex to recombination foci. Genetic and cell biological assays go on to compare the phenotypes of rmif-2 and rmh-1 and reveal both similarities and differences. Most strikingly, analyses of cytological markers of crossovers reveal that unlike other members of the BTR complex, including rmif-2, MSH-5 foci are completely absent in the rmh-1 mutant; however, the COs generated in rmh-1 (and rmif-2) mutants are dependent on MSH-5 (and COSA-1). Further, the crossover landscape is altered in the mutants and both are important for restricting heterologous recombination, although rmh-1 mutants have a more severe phenotype than rmif-2 mutants with respect to heterologous recombination. Overall, this is an important study that identifies a new member of the BTR complex in a system that allows for in depth analysis of the meiotic recombination phenotype.

As detailed below, statistics need to be added for several of the analyses. In addition, the following should be addressed in a revised manuscript:

1. Please consider changing the title. I recommend something like:

Caenorhabditis elegans RMI2 (RMIF-2) and RMI1 (RMH-1) have both overlapping and distinct meiotic functions within the BTR recombination complex

2. In the author summary, please define heterologous recombination or remove to more generally tell the audience the significance.

3. Line 71: “crossing overs” should be “cross overs”

4. Line 110: Please add, “In mammalian cells, . . .”

5. Lines 124-125: the authors write “ . . ., indicating that rmif-2 functions not just as RMH-1 stabilizer for all its activities in the germline.” The way this is written implies to me that rmif-2 has additional functions, while the data indicates that it is rmh-1 that has additional functions.

6. Table 2: Please include statistics – I don’t know whether 40% is different than 50%, for example.

7. Line 152: The authors indicate that there is “robust” IP between RMH-1 and RMIF-2. Perhaps you can quantify how much of RMH-1 is coming down with RMIF-2 to substantiate this claim, based on the images shown, I am not convinced it is "robust".

8. In Figure legend 1, the authors say that late RMIF-2 pachytene foci are brighter than mid-pachytene. Based on the image, it is not clear to me whether the foci are brighter. Did the authors quantify intensity? Is there a statistical difference?

9. Figure 2: Is there a statistical difference in RAD-51 foci between the different mutants as indicated in the results section?

10. Line 230: The authors indicate that in the absence of RMIF-2, there is “none” RMI-1 in the insoluble nuclear fraction, please change to below levels of detection.

11. Figure 4: The significance of the paper rests on the differential phenotypes of rmi-1 and rmif-2. I realize that rmi-1 has been previously analyzed by this group, but it would be much easier for the reader if more of the analysis of rmi-1 was included so that it can be directly compared with rmif-2. This was particularly noticeable with respect to the analysis of localization of the different BTR complex members. I would also like to see the quantification of TOP-3 foci in the rmif-2 mutants in Fig4D.

12. Line 260 title: Please consider changing the subtitle.

13. Line 274: the sentence is awkward. “ . .. accumulation was delayed and fewer foci were observed . . .”

14. I found the analysis of rcq-1 distracting and would recommend either expanding to more clearly state the significance, or removing.

15. Did the authors analyze him-6; rmh-1? This would be important for comparing with the phenotype of rmif-2; him-6.

16. The recombination mapping needs statistics, including analysis of interference.

17. Line 417 problem with sentence.

Overall, I think this is an important study that with the addition of statistics and the inclusion of additional data comparing rmi-1 and rmif-2 would make a substantial contribution to our understanding of the role of different members of the BTR complex in meiotic recombination.

Reviewer #2: In their manuscript, the authors identified RMIF-2, a new member of the conserved BTR complex in C. elegans nematodes. The new protein does not have significant sequence homology to RMI2, a conserved member of the complex. However, it functionally behaves as a member of the complex. The authors then go on to define the contributions of RMIF-2 to the meiotic prophase functions of the BTR complex, and find that it has both overlapping and unique functions with other members of the complex.

The BTR complex is crucial for the maintenance of genomic integrity, DNA repair, and several key meiotic functions. In addition, the biochemical activities of the BTR complex, and some of its members, have been characterized extensively. However, the consequences of specific genetic perturbations are challenging to parse out. Some of the complexity stems from the fact that members of the complex play different roles when acting in the BTR complex and outside it. Many times, with opposite effects. Hence, the exact functional contributions of the complex and its members to the biological processes it is involved in are not well understood.

Insight in the worm BTR complex would be of interest to the worm meiosis community, as well as to the many scientists working on pathways that maintain genome integrity. The main contributions of the manuscript are the identification of RMIF-2, and a high-quality characterization of the consequence of its deletion on key meiotic processes. Frustratingly, however, the manuscript does little to shed light on the specific functions of the BTR complex and its members.

For this work to be a significant contribution to the field it would require further mechanistic insight into the function of the BTR complex. This could come in the form of better characterization of some of the unexpected genetic interactions presented. For example, the synthetic sterility of double mutant rmif-2 him-6; or the surprising effect of rmh-1 deletion on the localization of MSH-5. Ideally, this additional analysis could be synthesized to a model diagram that would tease out the different functions that the manuscript invoked. In addition, key issues relating to quantification of cytological data, to the co-IPs, and to statistical analysis have to be resolved.

Major points are listed below:

1. Lack of MSH-5 foci upon rmh-1 deletion (Fig. 6A): If true, that is perhaps the most interesting finding of this manuscript. However, it warrant further scrutiny. As it stand, it is in apparent contradiction to near complete co-dependency of RMH-1 and RMIF-2 for localization (Fig. 3). If RMIF-2 is required for RMH-1 localization, then why do they have different phenotypes with regard to MSH-5 localization? In addition, since many chiasmata do form in the rmh-1 mutant (Fig. 2A), the localization data would suggest these are MSH-5-independent events. That latter conclusion would be very interesting if true, but genetically, the chiasmata in rmh-1 mutants appear to be mostly MSH-5 dependent (Fig. S2A). Alternatively, MSH-5 can act without forming foci. In sum, this result has to be cleaned up. First of all verified: different tag/antibody for MSH-5 should be used, and evidence of the correct genotype (i.e., sequencing of the tag being used) should be provided. Assuming this result is corroborated, follow up is necessary to explore its implications; namely, do MSH-5-independent COs form in rmh-1 mutants?

2. The co-IPs suffer from some technical issues. In Fig.1 the are several bands that should’t be there in the untagged input lane. The IP band in the FLAG blot is also hard to see. I recommend this blot be repeated, and all non-specific bands be clearly labelled. A strain with only HA::RMH-1 would be a good control to include in this experiment as well. In Fig. 3C&F: the quantifications don’t seem to match the blots. The faint bands in ha::rmh-1; rmif-2 NS fraction (panel B) appears as almost zero in the quantification. Likewise for the rmh-1; rmif-2::ha in panel E - the faint band yields almost the same quantification as the untagged lane, although they appear different in the blot.

3. Figures 7 and 8 lack statistical analysis for significance. That is essential. In Fig. 7 the results are challenging to interpret; why are the number of events shown, rather than their fraction? I would recommend that Fig. 8 include a diagram of the genetic assay being used. This is not a commonly used assay and it is not trivial to follow.

4. Figure 4B: Immunofluorescence is not inherently quantitative since many of the steps involve non-linear amplification. Unless sufficient controls are added, quantitive comparisons of intensity between different genotypes should not be carried out. At the very least, the gonads from the two genotypes need to be imaged on the same slide, and even then, quantitative comparison should be taken with a grain of salt. Alternatively, the linearity of the intensity measurements should be addressed by measuring it in a condition of heterozygosity for the tag (it should go down by ~50%). In this specific case, an additional issue is that even if the quantitation is taken at face value, the main difference seems to be not between the average intensities, but between the seemingly bimodal distribution for ha::him-6 and the only dim foci for rmif-2 deletion. A related issue plays out in Fig. 6B. If the difference in ZHP-3 staining is indeed so dramatic, it should be addressed by some form of semi-quantitative imaging, and discussed further in the text.

5. Table 1: All proteins identified should be displayed. It is not clear from the text whether RMIF-2 is the 5th most abundant one or not. But either way, the results of this IP/mass- spec experiment should be shown in their entirety.

6. Table 2: As noted above, some of the genetic interactions shown here should be explored further, as they might supply mechanistic insight that is currently missing in the paper. Most notably, the near synthetic sterility between him-6 and rmif-2. At the minimum, other double mutant in the BTR complex should be analyzed, and the double mutant him-6 rmif-2 should be analyzed cytologically to address the source of this near sterility. Second, the result regarding rcq-1 (both here and in the text) is currently superficial and adds little to the conclusions. I would recommend removing it altogether, or alternatively, expanding and contextualizing it better. Finally, SD (or other indication of distribution) should be provided for the male percentage data.

Reviewer #3: The paper by Velkova et al identifies the C. elegans functional homolog of RMH-2 and describes its role in meiosis. The work is well done, and is convincing that RMIF-2 is indeed the functional homolog of RMH-2. Overall, a complex picture of the role of the BTR complex emerges. The complex is essential for chiasma formation, crossover regulation and preventing illegitimate recombination. RMIF-2 may not be completely essential for all of these functions, but the rmif-2 mutant is phenotypically close enough to rmh-1 to suggest it mostly is. The paper could benefit from some additional analysis that will address some unclear relationships between RMH-1 and RMIF-2 and the outcome of DSB repair as well as some textual/presentation changes, according to my comments below.

Major comments

1) Some statistical analysis is missing or just presented in the figure legend in a way that makes it hard to follow (table 2, Figure 2C, Figure 5, Figure 6AD, Figure 7, Figure 8 and Figure S3). The word “significant” is sometimes used without showing the p values. It’s important to add p values in the figures and tables, so it will be easily accessible to the reader. Since there are many comparisons to be made, it may be advisable to focus on the more relevant statistical comparison: 1) all mutants to wild type 2) double mutants to the representative single mutants, 3) rmf-1 to rmif-2.

2) Performing these statistical analyses will be needed to support some of the claims made. In addition, it will help to clear if rmif-2 and rmf-1 show complex genetic interaction (epistatic relationship (duplicate recessive epistasis) in some assays (emb, DAPI bodies #, % males?) and additive in others (brood size, RAD-51 foci)), or if the perceived additive interaction may not hold following statistical analysis (all recessive epistasis). If indeed after statistical analysis it is still clear that there is different genetic interaction between the mutants for different assays, the authors need to explain why they show epistatic relationship in some assays and additive in others, despite the assays reflecting similar meiotic processes. Thus far they just state that some of the functions are not interchangeable (line 323-348) but it’s not clear why we get a mixed bag and the relations to specific phenotypes.

3) I have some issues interpreting Figure 7 and Figure S3 and thus the conclusions driven from this figure. In ref 19 (Jagut et al 2016), rmh-1 mutants show no statistically significant elevation in DCO numbers, while here they may do (X4 elevation, Figure S3) but in the absence of statistical analysis we don’t know if this is significant. I addition to a statistical comparison of wild-type to mutant, it will be important to know if the n values in these assays can identify one missing crossover on one of the six chromosomes. In rmif-2 only a single crossover is missing (7 DAPI bodies), but if this is a random chromosome and only one chromosome is interrogated by SNPs, the expected crossover category should only drop by 1/6. If n values are low, this may not be detected. Running the observed data vs. the “expected” number of crossovers based on loss of one crossover per 6 chromosomes and the n values of the experiment, would be helpful.

4) Line 330-332: It is stated that in rmh-1; rmif-2 double mutate “six aberrant bodies that differed markedly from the well-shaped bivalents in the wild type” it’s begging to ask if these DAPI bodies are formed by NHEJ events. How many DAPI bodies are observed in rmh-1; rmif-2; ku mutants?

5) Line 330-332 and figure 2AB. Can the authors confirm in another way (e.g., staining for axis proteins) that aberrant DAPI bodies are observed in the rmh-1; rmif-2 double mutates but not rmif-2 single mutates? This is also related to Jagut eI (ref19 ) observing that rmh-1 mutants show abnormal bivalent structure- I think it will be important to examine if rmih-2 has any “abnormal bivalents”.

6) Assuming that reduction of crossovers is not observed (but could have been detected, point #2), it requires better discussion. If crossover frequencies are not reduced, why are there univalent? I don’t see an explicit explanation to this the discussion. How is their explanation of these observations with rmif-2 connected to how the phenotype of rmi-1 mutants is explained in ref 19, destabilization of chiasma after crossover are formed?

Minor comments

7) Line 99-101 “Loss of function of both Rmi1 and topoisomerase 3 leads to meiotic catastrophe, due to persistent joint molecules that are resistant to cleavage by resolvase” How can the nulls act synergistically if they are in the same complex? Maybe indicate that they also have separate function?

8) Line 252- I don’t see much difference between background foci of Top3 and background foci of other proteins in BTR mutants. However, Top3 residual staining in the rmif-2 mutants is described as “cytoplasmic aggregates”, when this word is not used to describe other residual/background staining. Are these really bigger, or show different characteristics that I can’t see? I think the word “aggregate” is a bit too loaded, unless there is a reason to call them so…

9) Line 260: I would not call it “expression profile” but “localization” since “expression” makes the reader expect to see mRNA/RT-PCR or western blot data.

10) Line 272- I don’t see how it is “contrary to our expectations “ in figure 2A: rmif-2 had ~7 bivalents and rmh-1 has ~8.5. That should translate to a reduction of 1 crossover site (5 crossovers) in rmif-2 and 2 in rmh-1 (3-4 crossovers) and this is very close to what is observed with COSA-1 at zone 7: rmif-2 had 5 crossovers/COSA-1 and rmh-1 has 3/COSA-1.

11) Line 277 “rmh-1 3 (± 1.4 SD)”, to “rmh-1 (3 ± 1.4 SD)”

12) Line 315-320 and Figure S2A. The experiment counting bivalent numbers in rmh-1; msh-5 (S2A) is interesting. I agree, that it shows that many bivalents in rmh-1 mutants are formed by class I crossover. Except, I would not say “largely” in “the joint structures seen in diakinesis were indeed largely dependent on MSH-5.”, since the phenotype of the double mutant is essentially in the middle between rmh-1 and msh-5 (10.8 is between 8.5 and 12). These data therefore shows that about half of the remaining physical attachments between chromosomes in rmh-1 mutants are not class I crossovers. It is unclear if these are other types of crossovers or NHEJ events. Can rmh-1; msh-5 ku mutants be analyzed for bivalent numbers to resolve the question if NHEJ is involved? Are the bivalents observed in rmh-1; msh-5 “abnormal bivalents”?

13) Line 327: “Strikingly, we observed that in the rmh-1; rmif-2 double mutant, embryonic lethality increased to 56%” reading this it sounds like the double mutant showed a more severe phenotype than both single mutants, while it actually did not. I suggest to remove “Strikingly” (because it is what is expected) and add something like “, levels similar to rmh-1 mutants”

14) Line 364:” For chromosome IV, the difference was less pronounced” I don’t think IV differences are significant (also related to comment #1-2). If so, it cannot be states that there is a difference at all.

15) Line 402-406: “In contrast, the reduced number of univalents in rmif-2 mutants might be due to the higher recombination rates in this background,…The extra crossovers in rmif-2 mutants might connect the univalents, which likely arise through the absence of a pro-crossover activity that is shared by RMH-1 and also seems to be lacking in him-6 mutants” I found this sentence hard to read. If the authors like to propose that DCO reduce number of bivalents they need to support this better. This is also related to comment #6.

16) Line 406-408: “Thus, the reduced number of univalents in rmif-2 mutants might be linked to their lower rate of embryonic death compared with rmh-1 mutants” I would phrase it the other way around, since the univalents cause the death.

17) Line 408-409: “fact, a degree of redundancy between RMH-1 and RMH-2 (both being RMI1 homologs) is indicated by the embryonic lethal phenotype of the double mutant” RMH-2 gets into the discussion out of the blue (I think the reader needs a reminder, since it was only mentioned lastly in the introduction). In addition, it is not clear to me if the double mutant rmh-1;rmh-2 is inviable (stated in ref 19), or that it is viable but produced dead embryos (here?)- which is true? Please clarify.

18) Line 419-423: the lack of MSH-5 localization in rmh-1 mutants, but the demand for its activity (DAPI bodies in rmh-1; msh-5) may simply imply that MSH-5 is localized to crossover sites in rmh-1, but the levels are too low to detect. I don’t know if that was what the last sentence means to say…

19) Line 422-423: “Moreover, the formation of joint chromosome structures in rmh-1 depends on msh-5, so a future challenge will be to elucidate this RMH-1-specific activity.” It partially depends.

20) line 428-429: I don’t see why different activity is required since throughout the paper, despite the overwhelming resemblance of phenotypes between rmh-1 and rmif-2, rmif-2 always seem to be very slightly better, suggesting some BTR active complex can form without rmif-2.

21) line 435-437: “The delayed formation and reduced numbers of MSH-5 and COSA-1 foci that we detected in rmif-2 mutants argue that extra crossovers arising through the lack of such activities are not marked by MSH-5 and are likely class II crossovers, which are usually resolved by non-canonical resolvases.” This implies that the # of MSH-5 and COSA-1 foci in rmh-1 mutants is lower than the numbers expected from counting # of DAPI bodies. In rmif-2 mutants there are 7 DAPI bodied= 5 crossovers, and there are 5 COSA-1 foci and 5 MSH-5 foci, so I don’t see the need to call for other crossover pathways. In my opinion the only support for class II crossovers may come from the SNP data in comparison to COSA-1/MSH localization.

22) line 444 “In contrast, the rate of heterologous recombination is lower in rmif-2 mutants than in him-6 or rmh-1 mutants” please add citation for heterologous recombination rates in him-6.

Figures

23) Figure 2A- It’s not clear which ones of the four most right genotypes are significantly different from each other. I would assume spo-11 and rmif-2;spo11 are not but spo11 and rmih-1;rmif-2 are…

24) Figure 3C and F- could the data points be shown?

25) Figure 4D- This was quantified using different zones then the rest of the data. Where are that zones in the germline? I see the description in the figure legend but maybe a cartoon/image like Figure 2C will help.

26) Figure 7A- please add n values (not only in present them in the text)

27) Is Figure 8B a single data point? If yes, I don’t believe that this can be presented in this way. It looks like 8B is the result of the calculation in the right-most column in 8A. If so, it just needs to be incorporated to figure 8A, which now can be presented as a table and not as a figure…

28) Table 1- please use RMIF-2 instead or in addition to Y104H12D.4

29) I would suggest adding a cartoon of RMIF-2 compared to mouse/human RMI2 so the overall size and the position of the OB-fold domain can be compared.

**Have all data underlying the figures and results presented in the manuscript been provided?**

Reviewer #1: Yes

Reviewer #2: Yes

Reviewer #3: Yes

PLOS authors have the option to publish the peer review history of their article (what does this mean?). If published, this will include your full peer review and any attached files.

Reviewer #1: No

Reviewer #2: No

Reviewer #3: No

---

## [Decision Letter · Decision Letter 1]

9 Jun 2021

Dear Dr. Jantsch,

Thank you very much for submitting your Research Article entitled 'Caenorhabditis elegans RMI2 functional homolog-2 (RMIF-2) and RMI1 (RMH-1) have both overlapping and distinct meiotic functions within the BTR complex' to PLOS Genetics.

The manuscript was fully evaluated at the editorial level and by independent peer reviewers. You will see that both Reviewers #1 and #3 support the publication of this work. However, Reviewer #2 has remaining concerns that we ask you address in a revised manuscript. While elucidating the mechanisms might be beyond the scope of this work, I see that the technical concerns raised by Reviewer #2 are addressable by minor textual revisions.  

We therefore ask you to modify the manuscript according to the review recommendations. Your revisions should address the specific points made by each reviewer.

[LINK]

Yours sincerely,

Yumi Kim

Guest Editor

PLOS Genetics

Gregory P. Copenhaver

Editor-in-Chief

PLOS Genetics

Reviewer's Responses to Questions

**Comments to the Authors:**

Reviewer #1: The revised manuscript by Velkova et al., describes the identification and functional characterization of the C. elegans ortholog of RMI2 (RMIF-2), a member of the BTR complex critical for recombination processing. The authors have done a good job addressing the reviewers’ comments, including addition of statistics and additional analyses of the rmif-2 rmh-1 double mutant. Overall, this is an important study that warrants publication in PlosGenetics.

I only have a couple of very minor changes:

1. Line 40: remove the last “the”

2. Line 276: I think you mean rmif-2 (not rmh-1).

3. Line 315: please add: “as” depicted

4. Line 353-354: For me the “mitotic” failure came out of nowhere – what is the evidence that it is both meiotic and mitotic failure? Please clarify.

5. Line 365: remove “the” meiotic DSB repair

Reviewer #2: I appreciate the improvements made in the manuscript, especially with regard to improved precision of the text, and some added statistical analysis. I also appreciate the added data regarding the unexpected (lack of) MSH-5 localization in the rmh-1 mutant. That said, my fundamental concerns about the manuscript remain, and I don’t find the manuscript ‘much-revised’. Most notably, I think the manuscript still lacks mechanistic insight into meiotic functions of the BTR complex, and into the specific contributions of RMIF-2 to those functions. That issue was not addressed to a significant level in this revision, and remains the main hurdle for publishing this paper. Several other more technical points:

While the clarifications on MSH-5 localization validate the technical aspects of the experiment, the issue still remains mysterious. As the results stand, they suggest that we need to examine what do MSH-5 foci mean at all, since MSH-5 seem to be able to act without making foci. (Related argument could be made regarding the faint and bright HIM-6 foci, the functional meaning of which remains unknown).

The western blots and their quantification remain confusing. For example, in Figure 1B, why is there such a difference between the HA levels in the two genotypes in the 10% input part? In Figure 3B-C, there is clearly a band in the HA blot in the ha::rmh-1 rmif-2 genotype, but it is quantified as zero.

There are still unsupported claims regrading the IF data (meaning unquantified or not-statistically validated). Claims relating to intensity of foci or signal should be more robustly quantified or removed. The authors’ rebuttal is correct - these are difficult experiments. But without more robust controls and quantifications, the text should be further clarified to indicate the impressionistic nature of the observations.

Reviewer #3: The authors addressed all my concerns

I have a couple of minor suggestions (not required)

Line 357- I suggest refraining from using the term “non-significant increase”, as if it’s not significant, it’s not an increase, it should just say that there is no effect/change/increase.

Figure 8B- I’m happy the significance was added with Fisher’s exact test, but it still looks strange to have a single data point in a graph (consider adding to table instead?). I believe the same issue is in 7B.

**Have all data underlying the figures and results presented in the manuscript been provided?**

Reviewer #1: Yes

Reviewer #2: Yes

Reviewer #3: Yes

PLOS authors have the option to publish the peer review history of their article (what does this mean?). If published, this will include your full peer review and any attached files.

Reviewer #1: No

Reviewer #2: No

Reviewer #3: No

---

## [Editor Report · Decision Letter 2]

11 Jun 2021

Dear Dr. Jantsch,

We are pleased to inform you that your manuscript entitled "Caenorhabditis elegans RMI2 functional homolog-2 (RMIF-2) and RMI1 (RMH-1) have both overlapping and distinct meiotic functions within the BTR complex" has been editorially accepted for publication in PLOS Genetics. Congratulations!

Yours sincerely,

Yumi Kim

Guest Editor

PLOS Genetics

Gregory P. Copenhaver

Editor-in-Chief

PLOS Genetics

Comments from the reviewers (if applicable):

**Data Deposition**

http://datadryad.org/submit?journalID=pgenetics&manu=PGENETICS-D-21-00186R2

**Press Queries**

---

## [Editor Report · Acceptance letter]

7 Jul 2021

PGENETICS-D-21-00186R2 

*Caenorhabditis elegans* RMI2 functional homolog-2 (RMIF-2) and RMI1 (RMH-1) have both overlapping and distinct meiotic functions within the BTR complex 

Dear Dr Jantsch, 

We are pleased to inform you that your manuscript entitled "*Caenorhabditis elegans* RMI2 functional homolog-2 (RMIF-2) and RMI1 (RMH-1) have both overlapping and distinct meiotic functions within the BTR complex" has been formally accepted for publication in PLOS Genetics! Your manuscript is now with our production department and you will be notified of the publication date in due course.

With kind regards,

Andrea Szabo

PLOS Genetics

On behalf of:
